# Thousands of reptile species threatened by under-regulated global trade

Benjamin M. Marshall [ID] [1], Colin Strine[1] & Alice C. Hughes [ID] [2,3✉]

Wildlife trade is a key driver of the biodiversity crisis. Unregulated, or under-regulated wildlife trade can lead to unsustainable exploitation of wild populations. International efforts to regulate wildlife mostly miss 'lower-value' species, such as those imported as pets, resulting in limited knowledge of trade in groups like reptiles. Here we generate a dataset on web-based private commercial trade of reptiles to highlight the scope of the global reptile trade. We find that over 35% of reptile species are traded online. Three quarters of this trade is in species that are not covered by international trade regulation. These species include numerous endangered or range-restricted species, especially hotspots within Asia. Approximately 90% of traded reptile species and half of traded individuals are captured from the wild. Exploitation can occur immediately after scientific description, leaving new endemic species especially vulnerable. Pronounced gaps in regulation imply trade is having unknown impacts on numerous threatened species. Gaps in monitoring demand a reconsideration of international reptile trade regulations. We suggest reversing the status-quo, requiring proof of sustainability before trade is permitted.

[1] School of Biology, Institute of Science, Suranaree University of Technology, Nakhon Ratchasima, Thailand. [2] Centre for Integrative Conservation, Xishuangbanna Tropical Botanical Garden, Chinese Academy of Sciences, Menglun, Mengla Yunnan 666303, PR China. [3] Center of Conservation Biology, Core Botanical Gardens, Chinese Academy of Sciences, Mengla 666303, China. ✉email: ach_conservation2@hotmail.com

Unsustainable human activity is driving a sixth mass extinction[1], an ever-widening biodiversity crisis driven by habitat loss, pollution, invasive organisms, climate change and wildlife trade[2]. Although awareness of the scale of biodiversity loss is growing, assessments of the wildlife trade remain incomplete, despite reports that direct wildlife exploitation is the second most damaging human activity to global biodiversity[3].

International cooperation to curtail damaging wildlife trade culminated in the creation of the Convention on International Trade in Endangered Species (CITES) in 1975. Within animals, the CITES regulations primarily regulate trade of commercially traded or charismatic species, only recently covering lesser-known species (e.g., pangolins-2016). Recent CITES meetings have highlighted that including large numbers of low-value species would be prohibitively expensive to enact[4]; thus, thousands of traded species remain largely unmonitored. Outside of internationally regulated and monitored species, the dynamics of legal wildlife trade are yet unknown at the global scale. Assessments focusing on a small subset of species or locations (often using variable methods) can fail to reveal the true extent of wildlife trade and thus potential impacts on traded species, especially within groups such as reptiles[5-7].

The disconnect between trade regulation and source population health potentially threatens thousands of bird and fish species for pet trade[8,9]. Without global assessments, we cannot be confident in similar assertions for reptiles, despite their popularity as pets[10] and vulnerability to increased demand for novel species[11-13]. Gaps in conservation assessments leave many reptile species with little or no population data[14], meaning that many species could be being traded despite risks to population viability, especially if sourced from wild populations. At least 21 species have had their entire wild populations harvested by collectors using species descriptions[5], and numerous other populations have suffered declines from over-collecting[6,15].

Here we expand upon data from existing trade databases with an online webscrape of reptile retailers to build a global assessment of the reptile trade. We reveal global trade dynamics by mapping traded species origins, exploring species endangerment status, and reporting the extent of wild capture. By investigating the delay between species' descriptions and their first appearance in the trade, we show that newly described species can be rapidly exploited. We ultimately illustrate how current regulations fail reptiles, demanding immediate reconsideration of the status-quo.

## Results

**The scale of trade.** To begin to understand the impacts of the wildlife trade we must know which species are traded and the extent to which trade targets wild individuals. Examining the overlap between trade demand, trade regulation, wild capture, and species vulnerability will reveal dynamics that potentially hinder reptile conservation.

Online trade has become a major component of global reptile sales. We determined species present in the online pet trade using text pattern matching. Our analysis included both a 'snapshot' of species currently present on reptile trading websites, and a longitudinal trend from the most species-rich website using a web-archive to view both current availability and change over time. We searched 151 reptile selling websites in five languages using 64,342 keywords that covered scientific and common names of 11,050 reptile species (a combination of Reptile Database and CITES names; see 'Methods'). From 23,970 web pages, we detected 303,403 keyword hits associated with a species of reptile, comprising 4029 unique keywords (3043 of which were scientific names). Due to synonymisations, the 4029 unique keywords translated into a list of 2754 species (Fig. 1). Estimates

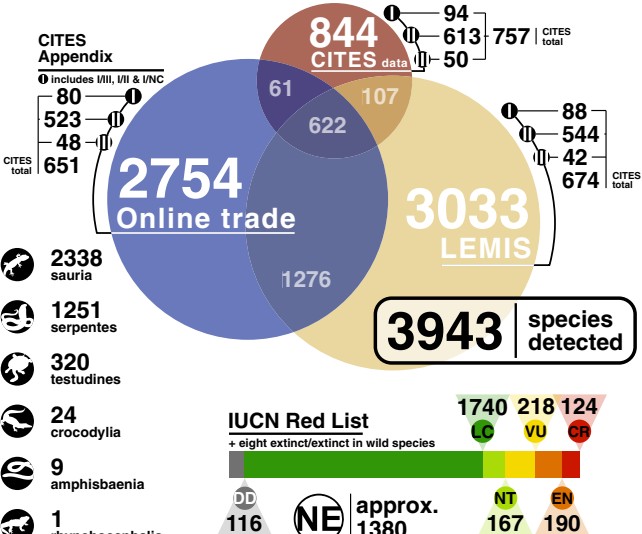

**Fig. 1 Summary of the number of reptile species detected in the trade between 2000 and 2019, and their status within CITES and the IUCN RedList.** Large numbers indicate the total in each of the three data sources, whereas the smaller numbers report the overlap between datasets (e.g., only 622 of the 3943 species were detected in all three datasets). CITES-appendix breakdown covers all potentially covered species, i.e., any historical name match indicating protection. Clade classifications are from Reptile Database listings. IUCN classifications: DD, data deficient; LC, least concern; NT, near threatened; VU, vulnerable; EN, endangered; CR, critically endangered. Species listed as lower risk/least concern are included in LC, species listed as lower risk/near threatened are included in NT. NE, not evaluated. NE species include those not evaluated by the IUCN and those we could not confidently match with a species name.

from sample-size and coverage-based rarefaction and extrapolation (all at 95% CI) suggested a total of $2296 \pm 61.13$ species (2188–2429) in the trade based on 2019 snapshot data (observed sample completeness = 0.947, 0.943–0.950; Supplementary Fig. 1 and 2), and $2936 \pm 46.73$ species (2854–3039) using the temporally sampled data (years 2002–2019, observed sample completeness = 0.965, 0.963–0.967; Supplementary Fig. 3). Online trade represents one of the interfaces between buyers and sellers, but the origin of these individuals is not declared due to a lack of regulations requiring such details. To supplement results from online trade we considered two major international trade databases where individual origin is recorded.

In considering international trade, the CITES trade database monitors all global international trade of CITES listed species (9% of reptile species), and the U.S. Fish and Wildlife Service Law Enforcement Management Information System (LEMIS) monitors the import of wildlife into the US, including listings of source (country, and if captive or wild) and destination, purpose as well as other information (i.e., legal trade, or from seizure). Both CITES and LEMIS trade databases include a percentage of seized items, but like online trade the majority of all items are legally traded (99%) and under 1% are from illegal trade (LEMIS 0.2% of individual reptiles are seized; and in CITES 0.2/0.4% (imports/exports)).

The overlap between online trade and international trade was substantial at 1898 species for LEMIS and 683 for CITES. Only a small percentage of these internationally traded individuals are non-commercial (1.54% in LEMIS representing 459 species). Both LEMIS and the online search results reveal most traded species are not CITES listed (76% online trade; 78% LEMIS), with similar percentages listed under appendix-2 that require export

permits to trade (19% 523/2754 online trade; 18% 544/3033 LEMIS), and appendix-1 that only permits non-commercial trade (3% 80/2754 online trade; 3% 88/3033 LEMIS). This contrasts to the CITES trade database, where 90% of species listed in trade are in a CITES appendix (e.g., appendix-2: 72.6% 613/844; appendix-1 11% 94/844). Not all CITES trade database species are covered by CITES appendices. Species reported in the CITES database may come from seizures (2.8–4%), or from shipments which include both species with CITES appendices, and unlisted species, in addition to other mechanisms.

The differences are due to CITES monitoring a smaller number of species that tend to be traded in industrial quantities for the fashion industry (e.g., reptile leather from crocodiles, pythons and monitor lizards). Online listings were for small numbers of individuals for pets. Listings in LEMIS covered large volume imports to fashion companies (importers are listed in the LEMIS database for each 'shipment'), as well as much smaller numbers of a diverse selection of species to other buyers.

**A growing problem?** We examined how species present on the most species-rich website (834 species) discovered in our 2019 snapshot changed between 2004 and 2018 (we excluded 2002, 2003 and 2019 because few archived pages were available, and the 2019 snapshot listings were sampled differently and thus not comparable, see 'Methods'), and compared the number of species detected in the trade databases (CITES and LEMIS; Fig. 2a). Overall raw counts of online traded species increased until 2008, reaching almost 1400 species in a single year, and have remained comparatively stable since, whereas CITES remains at a consistent 400–500 species annually, representing only about 29% of the species found online annually. When we control for online search effort (number of archived pages searched), we reveal a steady increase in the number of species traded on the most species-rich website (Fig. 2b). Every year since 2004 showed unique species not seen in previous or subsequent years (Fig. 2c). On average each year had 36.6 ± 6.69 unique species (35.7 ± 6.51 using only scientific name keywords). Raw counts of species indicate the consistent presence of CITES protected species in the online trade (75.5 ± 5.81; Fig. 2d).

We combined all three data sources, across all years, to produce an overall list of 3943 traded reptile species—36% of all reptile species.

**The threat from trade.** Though around 30% of reptile species have not been assessed for their IUCN RedList status, we can still explore the intersection of trade and threatened status. Based on the Reptile Database[16] names we show 5% (116/2563 species) of evaluated traded reptiles are Data Deficient, 68% Least Concern (1740/2563 species) and 21% (540/2563 species; Fig. 1) Vulnerable or worse.

Compared to online trade and LEMIS, there were fewer species belonging to each RedList category in CITES trade database apart from Critically Endangered (59 species total). Critically Endangered species are predominantly used commercially (95–96% of items), and dominated by a small number of species. For example, five genera in CITES account for 84% of traded items (*Alligator*, *Caiman*, *Python*, *Crocodylus*, *Varanus*) over the CITES 2004–2019 analysis, and all of these are predominantly traded for the fashion trade (reptile leather). On examining this further, based on the 'term' (i.e., what form the shipment takes, for example leather, skulls, carvings etc) most items (80–83%) were for fashion and other items included live (8–9%) food (6%), decorative (1%) and medicinal purposes (1%). The remainder were listed for other uses (1%). On a global basis 13 species were imported into over 125 countries, which were all crocodiles,

pythons and monitor lizards, and three species (two crocodiles and one python species) were exported from 125 countries. The commercial focus of CITES is further reflected in the regulation of fashion targeted species: 100% of crocodiles and 52% of Testudines described globally have a CITES appendix, compared with only 9% of lizard species and 4% of snakes. This coincides with species used commercially in fashion, with 20 of the 22 species of crocodilian in trade in the CITES database listed with commercial or personal 'purposes'.

**Origin of traded reptiles.** Quantifying the number of traded species, particularly vulnerable species, is the first step in understanding the wildlife trade's impacts. Yet the number of species impacted in itself does not demonstrate impact on wildlife populations. It is also crucial to understand the origin of traded individuals to assess if trade represents a genuine threat or if individuals are sustainably sourced. Both LEMIS and CITES list the origin of traded wildlife, thus providing insight into how international trade may impact the vulnerability of wild populations. In both international databases the majority of individuals may be sourced from wild populations.

In total, 53% of CITES reptile items traded are from wild-caught animals (whereas 33/36% are from captivity (imported/exported reported items (I/E)), this remains consistent if filtered to represent individuals at 48/47% wild (I/E) and 34/39% captive-bred I/E. The purpose is largely listed as commercial (95/96% of items I/E), with a minority for personal use (0.1%), highlighting that within CITES-monitored data, the majority is for commercial (largely fashion) purposes, and may overlook trade for other purposes such as the pet trade.

Data from LEMIS show that 92% of species have wild-caught individuals imported, and only 44% of species had captive-bred individuals imported. LEMIS lists 58.05% (58.08% excluding seized shipments) of individuals as wild-sourced (14,933,888/25,724,631), and 41.2% as originating from captive, ranching, or commercial breeding (10,605,330/25,724,631). For live individuals, 59.2% are wild-sourced (11,959,100/20,188,283), and the only type of item predominantly sourced from captive operations are live eggs (13.1% wild-sourced, 309/2367). There are notable differences in rates of wild capture between clades. Trade in Crocodylia is dominated by non-wild sources (90.6% non-wild), whereas Sauria (68.6% wild), Serpentes (46.6% wild) and Testudine (52.3% wild) individuals are more frequently wild-sourced (Supplementary Fig. 4). For the three extensively imported groups sourcing varied dramatically, with only 9% of individual crocodiles being wild-sourced relative to 76.5% of monitor lizards and 31.8% of pythons. Examination of RedList status (for species that could be connected to a RedList status) reveals that 5.17% (99/1,914) of LEMIS traded species are Critically Endangered, and a further 7.11% are Endangered (136/1914), illustrating the likely overlap of wild capture and vulnerability.

Though traded species diversity is high across the tropics (Fig. 3a), Vietnam is a major source of some of the more threatened species currently traded (Fig. 3c, d). In terms of proportion of species found for sale online, Europe, North America, and species-poor areas have the majority of their reptile species for sale online (Supplementary Fig. 5), whereas South America exploits a comparatively low proportion of their local species (Fig. 3b). However, the percentage of species in trade must be viewed in the context of the actual species richness, where South America (despite the low percentage traded) surpasses Europe and North America in absolute species. By absolute numbers, the Malay-Peninsula and Northern Vietnam are hotspots of traded species (Fig. 3a). The maps clearly illustrate

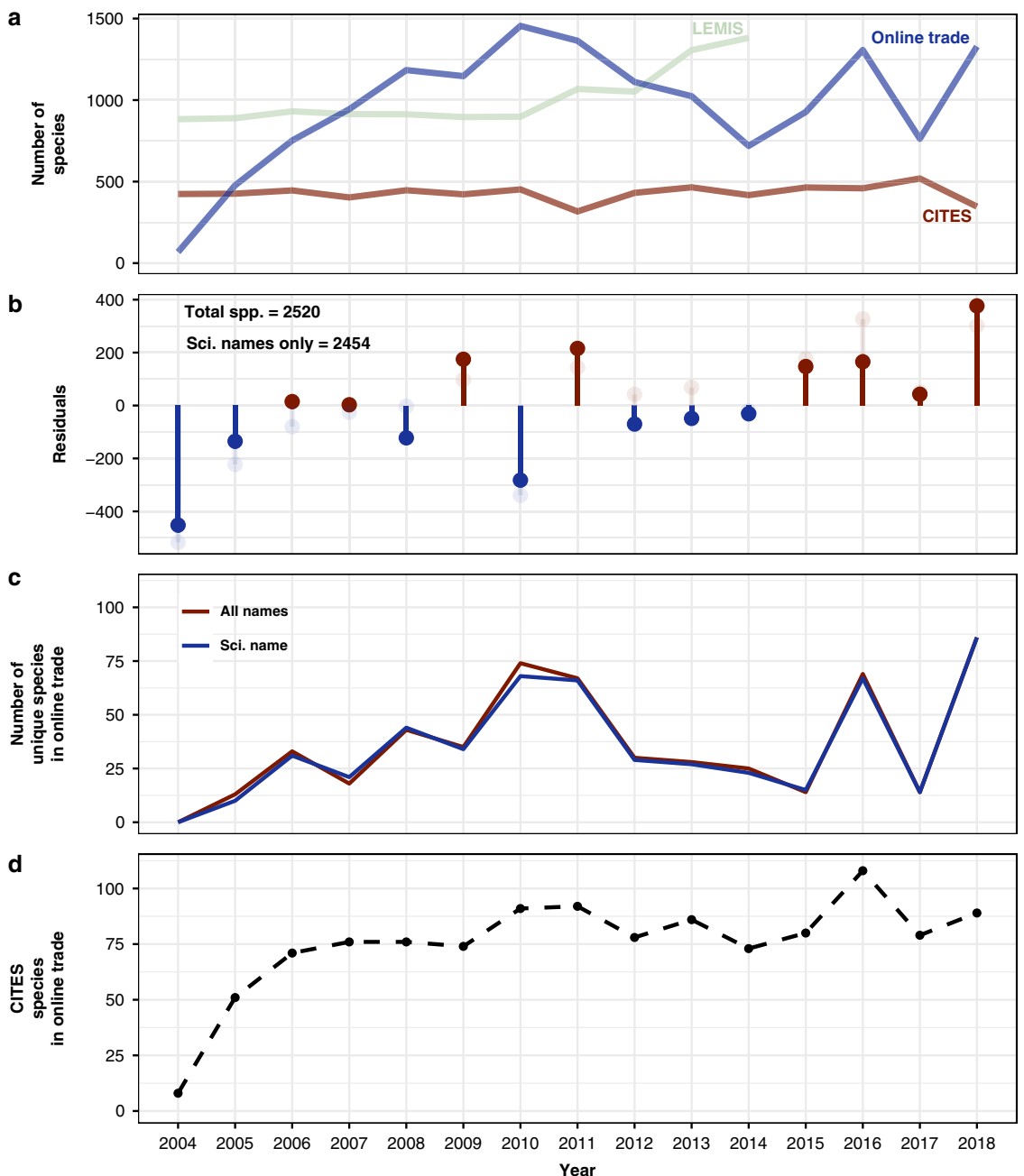

**Fig. 2 Number and uniqueness of species detected during 2004–2018. a** The raw species counts from CITES trade database, LEMIS and detected on the top reptile selling website (all keywords). **b** Trend in number of species detected on the most species-rich reptile trading website. Solid lollipops show residuals from a linear relationship between the number of pages available in a year and the number of species detected using all keywords. Light coloured lollipops show the residual species counts each year detected by only scientific names. **c** The number of species detected on the most species-rich reptile trading website unique to each year in the online trade. The red line shows the unique species detected using all keywords, while the blue line shows the species detected by scientific name only. **d** The number of species detected on the most species-rich reptile trading website listed in any CITES appendix in that year. Whether a species was included in a CITES appendix was determined by exact matching of Reptile Database name with CITES listed name.

the global reach of the reptile trade, highlighting near complete exploitation of species in North America and Europe and areas of high, but not total, exploitation in the tropics at around 50% of species present. However, these patterns are reinforced when the online and LEMIS sources are considered (Supplementary Fig. 5); CITES lists a maximum of 40 species in trade from any area whereas LEMIS and Online sources more than double this number (LEMIS 108, Online 98).

Almost no country (excluding depauperate islands, i.e., Iceland, Kiribati, Reunion) has more than 50% of their traded reptile species covered by CITES regulations, with the exception of Madagascar and New Zealand (63%, 53% of species in trade listed: Supplementary Fig. 6a, b). The highest proportions and number of Endangered, Critically endangered and Data Deficient species in trade exist in China, Madagascar, Vietnam and several Southeast Asian countries. Asia stands out as most at risk from

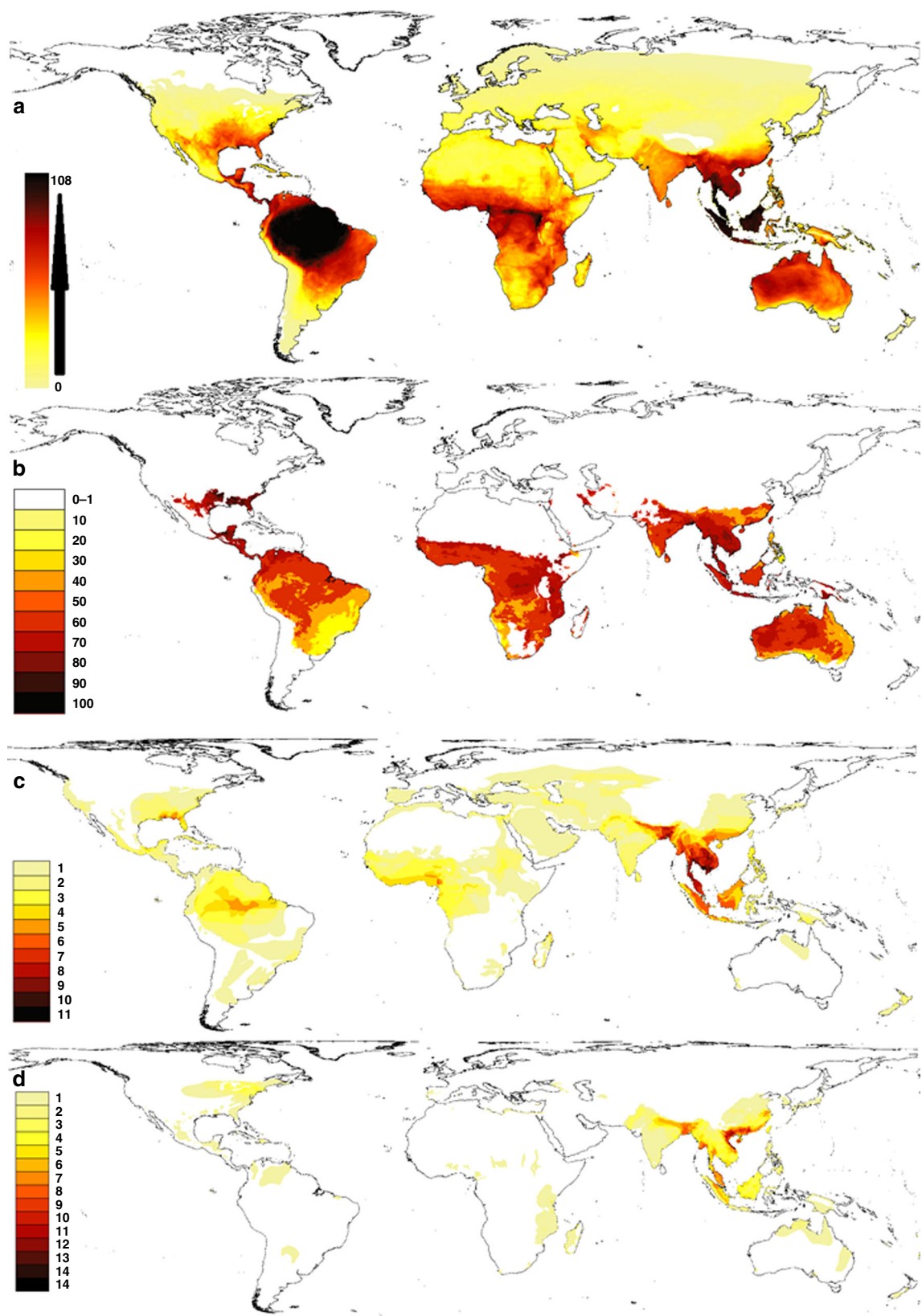

the trade in Endangered species (Supplementary Fig. 6c), whereas Africa has the greatest proportion of species in trade unassessed.

**The pursuit of novelty**. One major argument about banning the trade in any given item is that rarity and difficulty in procurement may inflate demand and value of an item, inadvertently fuelling trade. Yet despite the lack of regulations on the international trade for over 90% of known reptile species, novelty (newly described or rare) species may still be in high demand, and thus especially vulnerable. In total, 5.5% (133/2437) of species formally described after 1999 are already in trade. Ninety-two of these

**Fig. 3 Maps showing the species distributions of all traded reptile species based on species listed in trade in the three data sources (Online, LEMIS and CITES).** Reptile range maps of traded species are from the GARD database. **a** Number of traded species from any given location. Supplementary Fig. 5 shows the number of species listed in trade in each of the three sources separately. **b** Percentage of species in trade where over 50 species exist. Yellow colours indicate under 50% and red-black over 50% (areas with low diversity may have a high percentage of species in trade based on low numbers of species in total only diverse areas are shown here, whereas Fig. S5 shows percentage of species in trade for all species richness levels). **c** Origin of traded species listed as Endangered or Critically Endangered by the IUCN showing the number of species with that status in any given location. **d** Origin of traded species listed as Vulnerable by the IUCN and the number of Vulnerable species.

species could be connected (from the 2000–2018 timeframe) to a year of first appearance (Fig. 4a), while the other 41 species were only detected in the 2019 snapshot data (Fig. 4b). The unequal sampling between 2019 and previous years led us to treat species only detected in the 2019 snapshot separately. Species only detected in 2019 likely have an earlier initial date of appearance (missed due differences in sampling methods); including lag times based on 2019 detections would have biased the mean upwards. The true number of newly described species in trade likely is much greater than 133, as splits of species complexes are likely being traded under older names. For the species connected to a year of first appearance, there was a mean lag time of 8.12 (±0.58) years between species description and their appearance in the trade (Fig. 4c)—11 newly described species were detected in the trade by the year following description.

More recently described species occupy smaller ranges[17] (Supplementary Fig. 7), thus trade poses a considerable risk to species survival. The desire for rare or new species could lead to the targeting of species with progressively smaller ranges. Three percent of species traded have ranges of under 25 km$^2$, at least 9% have ranges under 1000 km$^2$ (based on GARD ranges), and many species are too poorly known or occupy too small an area to have their ranges mapped. Several genera listed here (*Cnemapsis*, *Cyrtodactylus*, *Goniurosaurus*) are frequently site endemic species, which may occupy just a single hill, yet many are traded within a decade of description. The trade of small ranging species with no baseline and no regulation is undoubtedly a threat to those species.

**Varying markets**. For online trade in 2019, we examined whether species or genera were uniquely for sale in specific languages (Supplementary Fig. 8). 758 (42.5%) of the species were present on websites using only one of the five languages, and at a generic level (excluding genera with 1 representative) English sites had 29 unique genera, German 23, Spanish and Japanese 4 and French 3.

Websites hosted 71.8 (±8.59; Supplementary Fig. 9) species on average, with species appearing on 6.24 (±0.25; Fig. S10) different sites. Fifty-nine websites stocked reptiles not found on any other website; 583 species appeared on a single site (33.5% of all traded species). The top two most species-rich sites hosted classified advertisements, and a further three classified advertisement sites appeared in the top 15 most species-rich sites. Other species-rich sites were large professional retailers, two of which prominently advertised wholesale. It is important to note that many other websites exist that were not assessed, some of which function in different languages using different species common names or species abbreviations. Thus, our numbers even at 36% of all species still only reflect a portion of global trade.

## Discussion

**Under-documentation, cause for concern?** Our findings suggest a minimum of 36% of reptile species are being traded, many are coming from wild populations, newly discovered species can be swiftly exploited, and a minimum of 79% of traded species are not

subject to CITES trade regulation. Particularly concerning is the convergence of vulnerability and desirability of newly described, small-ranged species. When presented together, our findings reveal a worrying situation where a huge number of reptile species are being exploited, with little international regulation, implying a lack of reliable a priori estimates of the impact on wild populations.

CITES aims to ensure that wildlife trade is sustainable, yet it largely focuses on only the most economically valuable species traded in large volumes, leaving species which may have niche markets, are lesser-known, or range-limited, unprotected and vulnerable to trade. Key differences exist between CITES listed species being traded under CITES monitoring, and those for sale online or documented as traded via LEMIS. For CITES data, the majority comprises a small number of species, traded in high volumes for the fashion trade; those traded online were almost exclusively for the pet trade; LEMIS had both high quantities imported for fashion and a huge number of species sold in small numbers for personal or commercial use, though there is also a large medicinal market including over 284 reptile species which we did not explore here[18].

The major reptile consumer markets are within Europe and North America, where captive breeding and regional trade could decrease the threat posed by pet trade to many species[19]. But without considerable improvement to captive breeding documentation, legal trade of captive-bred individuals can still enable wild collection[20,21], via the laundering of animals through legally sanctioned farms[22] or non-range countries[23], and potentially even bolster demand[19]. As a consequence of inconsistent or inaccurate metadata supplied alongside reptile sales online, it was impossible to assess what proportion of such species came from the wild. But LEMIS data indicated that 92% of traded species include wild collected specimens, totalling 58.1% of all individuals. CITES already demonstrates a considerable trade of wild-caught individuals across taxa at around 47–48%[24], and while this is apparently decreasing, it may principally result from captive breeding of the most exported taxa (such as crocodiles as shown here). There is no legal requirement to supply captive breeding evidence or mechanism to prove provenance for non-CITES listed species. Requirements to provide information on origin and origin state (captive or wild, as illustrated by LEMIS) would greatly assist with quantifying sustainable trade, but would require verification to avoid laundering. Although studies have found that captive breeding and ranching can provide alternative livelihoods and thereby enable conservation, for this to be achieved mechanisms to prevent laundering are needed, and the cost of rearing animals cannot substantially exceed that for collecting from the wild[25]. The potential for laundering where insufficient scrutiny exists is illustrated for the Tokay Gecko (*Gekko gecko*), for which Indonesia has an annual CITES export quota of 3 million. Captive breeding was claimed, but given the minimum resources and staff to breed that number of geckos[26], it was deemed, almost certainly, that the majority of individuals were wild caught.

Our data indicate the alarming scope of reptile trade, but are likely not comprehensive. Even using synonyms (average of

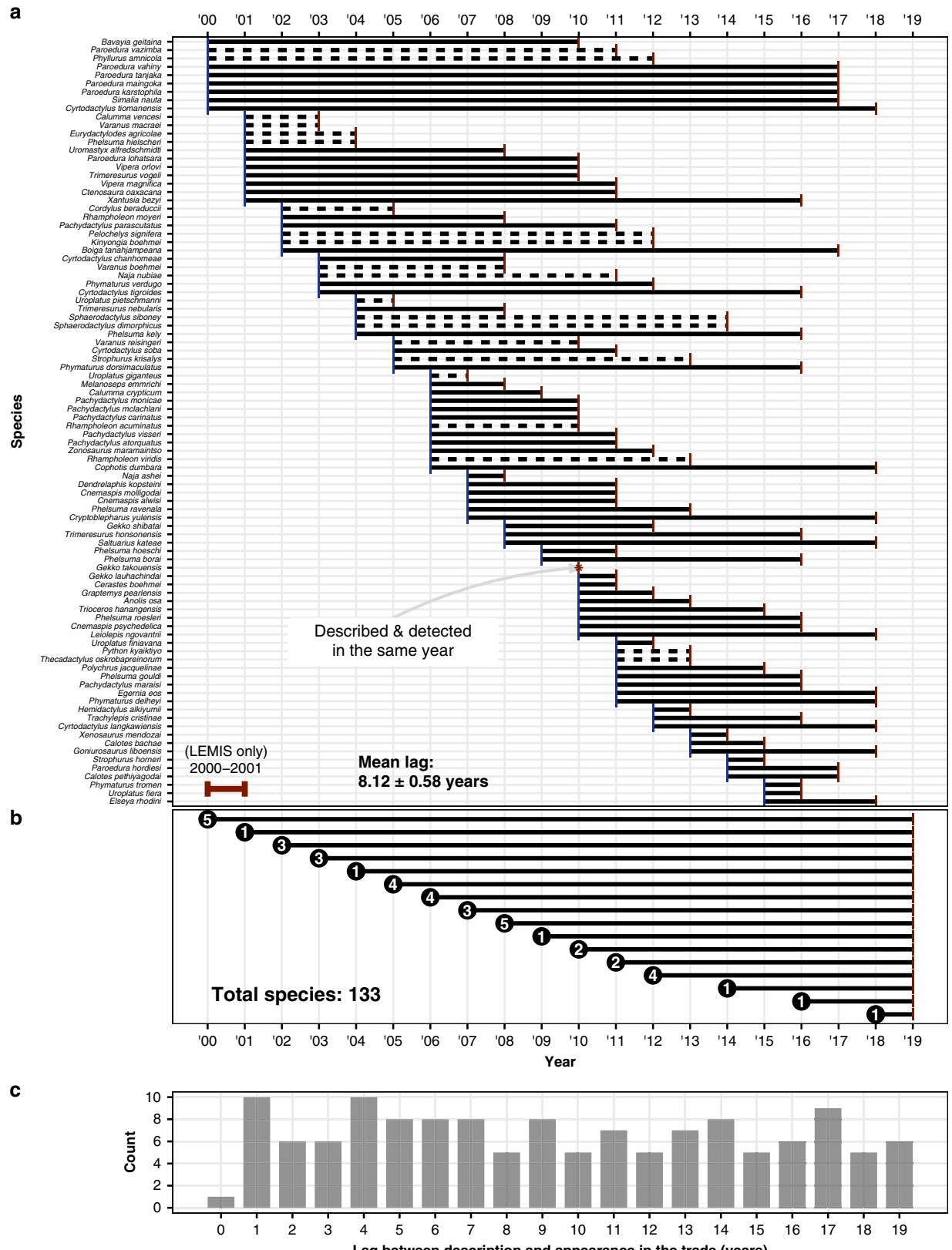

**Fig. 4 Time lag between post-1999 species descriptions and their appearance in the online trade or in the LEMIS database. a** Species detected during the temporal online sampling from the most species-rich website (solid lines), or in the LEMIS data (dashed lines) that allowed for an earliest-year-detected. The period 2000–2001 is only covered by LEMIS data. LEMIS data shown here excludes non-commercially traded species. **b** Counts of species described after 1999 but only detected in the 2019 snapshot data: the actual appearance date in the trade is likely before 2019. **c** Frequency plot of species lag times.

5.82 ± 0.06 per species), we detected 583 species (34%) on a single site indicating that further searches of more sites would likely reveal more species. If more languages (five were included in this analysis; see Fig. S7) were considered, or access to seller's stocklists was improved (16 potential websites had measures to prevent automated data collection) we would likely detect more species on retail websites (Supplementary Discussion 1). Further, social media enables wildlife trade[6] and in parts of the tropics is still a major component of reptile trade[27,28]; an extensive review of private online reptile groups would reveal more traded species, augmenting the already high numbers reported here.

**Reassessing reptile trade practices.** The scale of the trade and limited knowledge of the direct impact on wild populations justifies reassessing how we regulate international reptile trade. The USA instigated the Lacey Act to prevent animal trafficking in 1900[29]. While the Lacey Act preceded CITES, it expanded to recognise CITES listed species, in addition to species with local regulations on trade and export. This safeguard does not prohibit all domestic sales, but could limit import of threatened species, complementing CITES in cases where potential trade impacts are unassessed. Alongside implementing global LEMIS-like standards for wildlife imports (e.g., recording species, origin and purpose), tighter links between monitoring bodies would dramatically improve our understanding of, and ability to manage trade, especially if all listings require species, not genus, level identification.

While an expansion of a Lacey Act based body of regulations would somewhat mitigate the impacts on non-CITES assessed species, deficiencies in RedList assessments for tropical reptiles[14] also need addressing to prevent over-exploitation of reptile species from wild collection[30]. Without accurate and replicable population assessments, attempts to gauge trade impacts will be ineffective. Moratoriums on export of species from their native range, requirements for third-party verification of identity (i.e., designated centres within each country) and certificates for captive breeding in non-range states could also better safeguard difficult-to-study native populations. Trade under a precautionary scenario could be governed by an approved-list of tradable species with adequate population data to ensure trade does not pose a major risk to their survival. The reversal of current regulations to a precautionary system, where no-trade is the default, would relieve pressure to keep pace with taxonomic changes or descriptions, as new species would automatically be protected from trade.

Studies have demonstrated that CITES is consistently behind the IUCN in species assessment and inclusion[31]. Until population impacts are known and assessments complete, trade-bans from specific regions, where exports include threatened species, should be considered. For low-value species banning trade from key-regions may not drive trade 'underground' as can happen with high-value species[11], especially when such actions are used to stimulate regulated markets based on captive breeding as was found effective with crocodiles[32]. Conservationists actively hindered implementing the trade ban for birds on such grounds[33], but when eventually applied within Europe global bird trade decreased by 90%[34], in part because of the availability of non-wild-sourced alternatives for captive breeding[35]. With just under 4000 reptile species found to be traded within this study, there is ample stock already in captivity to justify the development of certified and monitored captive breeding within free-trade zones and prevent the need for commercial import. Such actions have already been used in the case of other taxa; for example, 'The Wild Bird Act' and 'EU Wild bird ban' have prevented the importation of exotic birds into the US since 1992, and Europe since 2005

(https://www.fws.gov/le/USStatutes/WBCA.pdf). In both cases, disease risk and impact on native birds was the stated case for regional bans[36], and these same justifications exist in the case of reptiles[37,38]. Yet though undoubtedly effective in reducing global trade, unintended consequences such as shifting routes and markets[35] also demonstrate a more holistic approach is needed. Such an approach would require CITES listing for any international export of reptiles, though species listed as Least Concern could be traded more widely if mechanisms for species identity verification existed prior to export. To counter disease spread and facilitate a deeper understanding of trade dynamics, systems like LEMIS should become global standards for the export of live animals, and especially for wildlife export. While many conservationists and organisations may challenge such an approach, as occurred in the case of birds[33], our data highlight thousands of species impacted by wild capture, including many which are new to science. By regulating what can rather than what cannot be traded internationally, we can considerably reduce the pressures on wild reptile populations.

**Are we failing reptiles?** Currently, CITES fails to adequately safeguard numerous reptile species: 36% of reptile species are in trade, four times more species are than the 9% monitored via CITES, and when mapped the maximum number of species in trade online or listed in LEMIS is more than double the maximum listed in trade through CITES. LEMIS highlights that the lack of regulations enables the legal import of wild-caught individuals, including both endangered species and endemic species occupying miniscule ranges <25 km². With a market motivated by novelty, and actively using species descriptions to locate and capture species (5.5% of species described since 2000 currently traded, and up to 11 species within a year of description), protecting unassessed species must become the default. The burden of proof should be shifted to demonstrate sustainability before species or populations can be traded. Better approaches are needed for the pet trade, where low financial values are unlikely to raise sufficient attention to uplist them to a formal CITES appendix. If we fail to mitigate the impacts of unregulated, but legal trade, small-ranged and endemic species may be the next victims of the ongoing biodiversity crisis.

## Methods

**Website sampling.** We used five different search terms, all translations of 'reptiles for sale' (reptile à vendre, reptilien zu verkaufen, 爬虫類の販売, reptil para la venta), on the appropriately localized version of two search engines (Google: https://www.google.com/, https://www.google.fr/, https://www.google.de/, https://www.google.jp/, https://www.google.es/ and Bing: https://www.bing.com/?cc=en, https://www.bing.com/?cc=fr, https://www.bing.com/?cc=de, https://www.bing.com/?cc=jp, https://www.bing.com/?cc=es) to retrieve a list of reptile selling websites, extracting URLs using the XML v.3.99.0.3[39], assertthat v.0.2.1[40] and stringr v.1.4.0 packages[41] in R v.3.5.3[42] and R studio v.1.2.1335[43] (Supplementary Code 1). We completed searches in Firefox[44], while signed out of search engine accounts and in a private window to minimise browsing history's impact on searches.

Once we had generated a list of search result URLs, we manually reviewed each URL's content—679 websites led to 151 searchable reptile selling websites. Our review had three goals: ensure the website was selling reptiles, check whether the website terms and conditions did not explicitly forbid automated data collection, and identify the most appropriate method of searching the content of the website (see Supplementary Data 1 for example of review datasheet).

We employed a hierarchy of five search methods depending on the structure of the website and the display of stocklists (Supplementary Code 3). The hierarchical approach minimised the analysis of irrelevant pages and minimised server load.

(1) Searched only a single html or pdf page. Where the seller has supplied a full list of stock, we could review all animals sold with only a single page using the downloadr v.0.4 package[45]. Occasionally animals were listed on large pages that displayed a store's full non-animal stock as well. For stocklists supplied as a pdf, we manually downloaded them and accessed the text using the pdftools v.2.2 package[46]. (2) Systematic cycling through search results. Forty-nine websites with adequate search functions allowed us to request all reptiles for sale, then examine the pages

of search results one-by-one. We employed this method when website search results contained the complete details of the reptiles for sale on the search page. We ceased cycling through search pages when a URL returned a 404 error, or when 100 pages had been cycled through. Hundred pages were surveyed to prevent endless cycling back onto initial pages, or deriving errors from misinterpreting the number of search pages returned, while still exceeding the number of pages on most sites. We performed a post hoc review of ten sites searched using a cycle search method to check whether species ordering could have led systematic biases for names near the beginning of the alphabet or price. For four websites we could not determine how species were ordered, for six websites species listings were ordered by date, and for one website, species were ordered by popularity. Thus even for sites with more pages, we feel the results will not be impacted by biases given the inconsistency of approach for ordering entries on different sites. The 100-page limit may have led to missing species on large websites, but undercounting likely only affected a small portion of the websites searched via cycling methods and overlap between websites species lists mitigate suboptimal sampling on any particular website (see species-accumulation curves, Supplementary Figs. 1 and 2).

(3) Systematic cycling followed by level 1 crawl. We employed this method when sites had adequate search functionality but the details or full names of species for sale were buried one level deeper into the search results. In these instances, we ran a level 1 crawl on every search results page (Rcrawler v.0.1.9.1 package[47]). We followed the same stop criteria as the single page systematic cycling, 404 error or 100 search result pages.

(4) Basic level 1 crawl. Some sites had a full species list but split between clades or categories. In this case we passed the page containing links to all the different clade lists and completed a level 1 crawl[40].

(5) Basic level 2 crawl. We required a level 2 crawl[47] when the subsection of reptiles for sale was more specific. For example, to detect a *Boa constrictor* on a site that divided its stock multiple times would require moving from the 'snake' section, through to the '*Boa*' section where the details of the stock are listed.

We employed these five search methods hierarchically, 1–5, and included 20 s delays between crawled requests to minimise server load on reptile selling websites. For search method 3, there was a significant chance of duplicated pages being returned; we removed duplicated pages prior to keyword searching. A few sites required multiple methods to extract complete stocklists. For search methods 3, 4 and 5, we limited the crawl further by selecting (where possible) keywords within a website's URL that must be included for a page to be searched. For example, a website may list animals on pages that all include the pattern '/category=reptiles/', therefore limiting the search of irrelevant non-stock pages.

We augmented our 2019 snapshot sampling by exploring the archived web pages stored on the Internet Archive[48]. For the most species-rich site (from the 2019 snapshot) we retrieved all archived web pages using the Internet Archive's Wayback machine API[49], adapting code from the wayback v.0.4.0 package[50], with functions from httr v.1.4.1[51], jsonlite v.1.6.1[52], downloader v.0.4[45], lubridate v.1.7.4[53] and tibble v.2.1.3 packages[54]. We limited the search to pages directly pertaining to sales.

Though our online search analysis provided the number of mentions per species per page, we do not detail these numbers because sellers may list multiple individuals at once, sellers may post the same advertisement numerous times, or that advertisements can be repeated on different pages within the same website. Therefore, numbers derived from online analysis did not provide a reliable estimate of numbers per species for sale, and we elected to restrict analysis to binary species appearances.

**Keyword generation**. We use the complete list of 11,050 reptiles created by Reptile Database[16], updated 14 August 2019, as our naming standard. We downloaded the complete list from http://www.reptile-database.org/data/, then fed the list of species into code designed to query and extract all common names, historic scientific names and locality information for each species from Reptile Database. The extraction code made use of functions from stringr v.1.4.0[41], XML v.3.99.0.3[39], xml2 v.1.2.2[55] and rvest v.0.3.5 packages[56]. We combined the resulting list with names, both common and scientific, supplied by CITES (http://checklist.cites.org/#/en [accessed 6 September 2019]) using the dplyr v.0.8.4 package[57] (Supplementary Code 2). Five CITES listed species had no matching counterpart in the Reptile Database; we determined that this was caused by minor spelling mistakes. We included both spellings in our complete list of species keywords. Overall, our species keyword list comprised all scientific and common names from both Reptile Database and CITES (Supplementary Data 2), with an average of $5.82 \pm 0.06$ s.e. (standard error) per species and grand total of 64,342 terms (s.e. calculated using the pracma v.2.2.5 package[58]). Common names were predominantly English, French, German or Spanish, but occasionally included local names. We compared the number of species detected via scientific names, to the number of species detected via a combination of scientific and common names because previous work highlighted that, while correlated, they can produce different search results[59].

**Keyword searching and species comparison**. On a site by site basis, we cleaned each page's html code of extraneous punctuation, numbers and spacing, replacing them with single spaces. That way, two-word keywords split by line breaks, punctuation or double-spacing appeared the same as those split only by spaces.

After cleaning the html code, we searched each page for keyword matches using the stringr v.1.4.0 package[41] (Supplementary Code 4, 5). Because of the large quantity of keywords and the high computational cost of collation string matching, we used a fixed string matching set to be case insensitive. Fixed string matching has the disadvantage of being sensitive to variation in how diacritics or ligatures are displayed as single or multiple characters. Our keyword searches returned the website, page number (as an index relating to the total number of pages retrieved from a given website), the keywords detected, and the corresponding Reptile Database name (Supplementary Data 3).

We searched the pages obtained from The Internet Archive[48] using the same list of species (Supplementary Code 4, 7). Because there is likely a connection between the number of pages available and the number of species detected, we regressed the number of species detected in a year against the number of pages searched ($n = 15$, intercept $= 483.72$, gradient $= 1.65$). We excluded 2002, 2003 and 2019 for this regression because they had considerably fewer pages than all other years (mean of $3.7 \pm 1.2$ pages, compared to mean of $296.6 \pm 48.8$ pages). We plotted the residuals from the regression alongside counts of unique species per year and the number of species included in CITES appendices. To show the sensitivity to the keywords used, we counted the number of unique species in two ways: (1) counting all species detected using either scientific or common name keywords, (2) counting species only detected using scientific name keywords. The two keyword groups produce slightly different yearly species lists; therefore, changing the number of unique species per year and yearly residuals.

We compared the list of species names generated by our keyword searches to those listed in CITES. Because names of species have changed, we first converted the CITES scientific names to the most recent Reptile Database used name. However, due to species synonymisations, splits and name changes, comparisons between the list of traded species and CITES species contain some ambiguity. The ambiguity can be seen in the variation between the number of traded species covered by CITES when comparing to only the top Reptile Database name, versus when comparing the CITES list to any historically used name of traded species. For general reporting we used the more generous matching using any historic name, boosting the estimations of CITES coverage. For examination of counts of CITES covered species traded over time (Fig. 2d) we used the more stringent single name matching because of the added complexity of a changing list of CITES species and the assumption that new CITES listings would use the most recently accepted name.

**Data exploration and display**. We used forcats v.0.4.0[60] and dplyr v.0.8.4[57] to manipulate data, and ggplot2 v.3.2.1[61], scico v.1.1.0[62], ggpubr v.0.2[63] and ggforce v.0.3.1[64] to generate the plots. We undertook keyword searching in R v.3.5.3[42] and R studio v.1.2.1335[43]. Silhouette images were obtained from http://phylopic.org/, in cases where the images were not public domain for free from attribution they were produced by Aline M. Ghilardi (CC BY-NC 3.0) and Roberto Díaz Sibaja (CC BY 3.0).

We explored the completeness of our samples—2019 snapshot and temporal—in two ways. The first was only applied to the snapshot data. We built an accumulation curve illustrating the relationship between the number of sites sampled and the number of species detected. We accomplished this by randomly resampling a subset of websites, increasing the subsample by one website until all were included. We repeated the resampling process 100 times, and plotted the results with a loess smoothed curve. The second method we applied to both snapshot and temporal data. Using the iNEXT v.2.0.19 package[65,66], treating our data as raw incidences, we calculated both sample-size and coverage-based rarefaction and extrapolation metrics providing us with estimates of total species richness and sample completeness. For snapshot data we used 'website' ($n = 151$) as the resampling method, for the temporal data we used 'year' (2002–2019, $n = 18$).

We compared our data to two international trade databases (compiled species list is available in Supplementary Data 5, code for review of data sources is available in Supplementary Code 9): CITES and United States Fish and Wildlife Service's Law Enforcement Management Information System (LEMIS). Following the online web scraping, the same types of analysis and cleaning were applied to all three databases. CITES data was retrieved from https://trade.cites.org/# on 13 May 2020) using the comparative tabulations for all 'reptilia' and the appropriate years (the snapshot of 2019, and 2004–2019) to download all reptile species traded over this time. We retrieved LEMIS data[67,68] (v.1.1.0) via R using the lemis package[69] (Supplementary Code 6). LEMIS data includes records of imports to the USA, alongside information pertaining to purpose, quantity, origin, date among other metadata, therefore quantitative data on imports for each species, or based on origins and source could be calculated in R using dplyr v.0.8.4[57]. As for the CITES species lists, the unstandardised LEMIS names were matched to those present in Reptile Database (operating as our backbone nomenclature), leading to both synonymisations and splits. A LEMIS name was converted to a Reptile Database name if it matched any current, common, or historically used name. Names would fail to match if misspelled. By LEMIS naming, there were 639 instances of genus level listing, that were matched to 510 Reptile Database names. Of the 510 converted names, 442 appeared in other sources, suggesting genus level listings in LEMIS did not inflate species counts. Outside of generic level listings, 83 full names could not be matched. We compared the 83 names to the traded list from other

sources, looking for names with fewer than 5 different characters (using the similiars v.0.1.0 package[70]); 56 species were found to be present in other sources by this metric. Those that failed to be converted were not included in total species counts; as final counts were entirely based on Reptile Database explicit species naming. Final species counts from all data sources are based on unique Reptile Database names and do not include any remaining generic identifiers after this synonymization/split process.

Though the research focused on the percentages of species vulnerable to trade based on various forms of IUCN and CITES categorisation, we made some efforts to quantify the proportions of items with different statuses within CITES and LEMIS. Quantifications were made using a number of different approaches. Online assessments were not directly quantified due to the possibility of listing the same individuals multiple times, or having mixed batches of specimens with variable numbers. For CITES we used the summary statistics tool in ArcMap 10.3 to quantify the means and totals for the numbers exported and imported (and listings of both are provided throughout where the numbers differ), and the range for each species or endangerment status is provided in text (or a single number if they were the same). RedList status was associated with the data by joining the scientific name field between the two databases. Sums were made for various sources, purposes and endangerment statuses for CITES data using this same approach, based on the 2004–2019 data from the CITES trade portal. 'Terms' (i.e., skins) were also explored, recategorising the standard terms (57 were used for reptiles) into nine (i.e., fashion, live, food, decorative, medicinal, specimen, egg, body, other uses), then summing the total item number imported and exported and determining the percentage. In addition to this we tried to quantify the trade in wild captured individuals within CITES. To try to represent individuals, terms from the CITES trade database were filtered to only include bodies, carapaces, eggs, live, shells, skeletons, skins, specimens, trophies, as most of these are mutually exclusive, though the huge quantity of reptile leather and meat could not be converted to representative individuals, skins or bodies listed as weights were also removed. Following from this the individuals from each source imported and exported could be calculated to percentages of individuals from the wild or captive bred within CITES, though these percentage values were very similar to summed total values showing the results are consistent. To investigate the extent of wild capture in LEMIS data, we restricted our summaries to items that represent individuals (whole dead bodies, live eggs, dead specimens, live individuals, full specimens, substantially whole skins, and full animal trophies), filtering out 75.6% other reptile items (79,812,310/105,536,941) leaving us 25,724,631 items to review source and purpose. The filter terms are close to those used in other recent publications which also quantified elements of trade ("live", "bodies", "skins", "gall bladder", "skulls", "heads", "tails", "trophies" and "skeletons")[71], but we also excluded body-parts that may have come from the same individual (i.e., skin and skull) which may otherwise inflate numbers (79,812,310 items including skulls and skeletons; 79,796,472 excluding skulls and skeletons). The filtering to individuals made negligible difference in summaries of origin (wild or captive): 58.17% wild-sourced without the filter, 58.05% wild-sourced with the filter (61,390,757/105,536,941 without; 14,933,888/25,724,631 with); 41.32% captive sourced without the filter, 41.23% captive sourced with the filter (43,611,039/105,536,941 without; 10,605,330/25,724,631 with). Our quantification of non-commercial trade was calculated by the number of individual animal items listed as Scientific, Reintroduction, or Biomedical research; our quantification of captive sourced trade was calculated by the number of individual animal items listed as being bred/born in captivity, commercially bred, or from ranching operations. We excluded all instances of NA in either purpose or source filters (127,881 reptile items had a missing source, purpose, or description). We additionally include clade-based analysis of source, as some taxa (i.e., crocodylia) may be more impacted for fashion trade and are imported in greater numbers. For clade-based summaries of wild capture, we summarised the quantity of traded items by genus, and further simplified the genus-summary to clade using Reptile Database genera and family information. For genera missing from Reptile Database (e.g., where genus information was family such as Varanidae), we manually assigned the clade.

Maps were created using the Global Assessment of Reptile Distributions (GARD) database[72] combined with each species list as appropriate using join field and then connecting by scientific names in ArcMap 10.3 based on the corrected lists. Join by field was also used to connect species to their RedList status (downloaded from https://www.iucnredlist.org/search) and CITES appendix (from http://checklist.cites.org/#/en). To create hotspot vulnerability maps we extracted each group with different IUCN classes then used 'count overlapping polygons' to count the number of species with each status in any given area. This was then repeated separately for the species listed within each of the three data sources, to map the species listed as traded within each separately in addition to the total number in trade.

To obtain overall number of species and percentage of species we separated each species polygon for species in trade, and all species using QGIS, then converted them to rasters with a resolution of ~1 km using ArcCatalog. Mosaic to new raster was then used in groups of 200 species, then all mosaics added to determine overall richness for reptiles, and richness for reptiles in trade, and the percentage of reptiles in trade determined using the raster calculator ((traded_species/all_species)*100). Other trends, i.e., the percentage of species coming from different sources or with different statuses was calculated in Excel using basic approaches to quantify listings with different qualities (i.e., seized, wild,

commercial and personal use) and the percentage with that status within CITES based on the number of exports and imports. For more extensive analysis of multiple factors, summary statistics were used in ArcMap after joining fields to connect species data from traded specimens of the three data sources with RedList assessments. This provided some simple statistics to further understand patterns as detailed in text, as CITES data lacks the detail of some other data sources; it was largely used to understand what species were in trade relative to existing regulations and threat.

To determine trends on a country basis we joined the CITES appendix and RedList status to the GARD layer. We used QGIS to separate a global country layer (http://thematicmapping.org/downloads/world_borders.php) into constituent countries, then clipped the GARD layer into each country with trade status noted. ISO2 codes were added to each of the country layers, then each country merged again to list each species and country and thus provide a species list for all countries within the GARD database. The number of species in and out of trade, and with and without an appendix was then calculated for each species using summary statistics, and this was repeated for each RedList status (Supplementary Data 6). For all species listed in Reptile Database but with no GARD layer, countries were listed separately and the process repeated based on the listing on the website, then the total combined with that from the GARD layer to map country richness, and the number of species with each trade status and endangerment to provide an understanding of the level of potential threat to the reptile faunas of different regions based on the trade, threat and CITES appendix of species listed in those countries.

For exploration of the time lag between species descriptions and their detection in the trade, we relied on the date of description from the author details supplied by Reptile Database. We extracted the earliest description date for each species using the stringr v.1.4.0 package[41] (Supplementary Code 8), and compared this to the year reported alongside the archived pages or trade date in LEMIS. For species detected only in the snapshot data we used 2019 as their date of appearance in the trade and did not include them in the calculation of mean lag time. We only included species that had been detected directly with the scientific name of the new descriptions, subsequent name changes or common names were ignored for this analysis. We also excluded species listed as only being traded for LEMIS non-commercial purposes in this part of analysis.

**Reporting summary**. Further information on research design is available in the Nature Research Reporting Summary linked to this article.

## Data availability
Original data generated from our online trade survey (Supplementary Data 1–4), alongside a compiled datasheet of traded species is available for download as supplementary material (Supplementary Data 5). Website names/URLs have been redacted to preserve their anonymity. We also include a datasheet of country-level summaries: species listed in each CITES appendix, unlisted in trade from each country, and IUCN RedList status (Supplementary Data 6). We have included data obtained from LEMIS, CITES, and Reptile Database in Supplementary Data 7.

## Code availability
R code for data curation, analysis and visualisation are available for download as supplementary material (Supplementary Software 1). Supplementary Code 1. Code used to extract search results URLs; Supplementary Code 2. Code used to compile and generate complete species search list; Supplementary Code 3. Code used to retrieve online trade website data; Supplementary Code 4. Code used to search websites html; Supplementary Code 5. Code used to compile and review online trade results; Supplementary Code 6. Code used to retrieve LEMIS data; Supplementary Code 7. Code used to retrieve archived web pages and examine the trend over time; Supplementary Code 8. Code used to examine the lag time between species description and their appearance in the trade; Supplementary Code 9. Code used to generate guides to Fig. 1 Venn diagram and further data review.

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

## Acknowledgements

We thank the Suranaree University of Technology (SUT) for providing the resources required to undertake this research. We would also like to thank Daniel Hughes, Charlotte Hughes, Aubrey Alamshah, Matt Crane and students from SUT for proofing earlier versions of the manuscript to ensure comprehensibility. Chinese National Natural Science Foundation (Grant #: U1602265, Mapping Karst Biodiversity in Yunnan). Supported by the High-End Foreign Experts Program of Yunnan Province (Grant #: Y9YN021B01, Yunnan Bioacoustic monitoring program). Supported by the CAS 135 program (No. 2017XTBG-T03). Priority Research Program of the Chinese Academy of Sciences (Grant No. XDA20050202).

## Author contributions

Conceptualization, A.C.H. and C.S.; formal analysis, B.M.M. and A.C.H.; investigation, B.M.M. and A.C.H.; methodology, B.M.M. and A.C.H.; visualization, B.M.M. and A.C.H.; writing—original draft, A.C.H., B.M.M. and C.S.; writing—review & editing, A.C.H., B.M.M. and C.S.; supervision, A.C.H.

## Competing interests

The authors declare no competing interests.
