## [Peer Review File · Nature Communications]

Reviewers' Comments:

Reviewer #1:

Remarks to the Author:

In this study, the authors have sought to characterize the legal wildlife trade in reptiles at a global scale. This is a worthy research goal, and the results are likely to be of broad interest to the conservation community. While the literature already contains various large-scale, primarily descriptive wildlife trade papers, I believe this manuscript distinguishes itself in a number of ways:

- 1) The authors have made a substantial effort to quantify the online trade in reptiles. This is a relatively novel, but critical, data source to consider, and the methods used for this portion of the study should be relevant to conservationists seeking to quantify the online wildlife trade across disparate taxa.
- 2) The authors have made the effort to integrate their novel online trade data with other notable existing wildlife trade data sources, namely the CITES Trade Database and the US-centric LEMIS dataset.
- 3) This manuscript addresses the short timeframe over which newly described reptile species may be impacted by the global wildlife trade. I think the authors are right to emphasize the importance of this particular result; this is a specific consequence of the wildlife trade that is rarely addressed in other studies, and the implications for rapid overexploitation of newly described species are alarming.

For the reasons above, I think this manuscript is deserving of eventual publication. At the moment, however, there are a number of outstanding issues that I believe are in need of improvement. Primarily, these are related to communication and interpretation of the study results. I have organized my specific feedback into major and minor comments (see below). I do not think any of my suggestions should require substantial data re-analysis, but addressing these issues will make for a more easily interpretable paper. Finally, I would note that there were relatively minor but somewhat pervasive grammatical issues (i.e., simple typos, problems with

sentence construction, etc.) that should be addressed prior to publication. Some of these I've highlighted in my minor comments, but that is not a comprehensive list.

Evan A. Eskew

Major Comments

Page 1, “Piecemeal assessments fail to reveal...”:

This seems a little vague. I think it’s important to emphasize what you view as the limitations of prior work since that helps to distinguish your current efforts. So, in what particular ways do you view these assessments as subpar? A focus on too few species? Inadequate data? A focus on limited geographic regions?

Page 1, “...only CITES trade portal includes a percentage of seized specimens...”:

This is not strictly true. While the vast majority of LEMIS data represents wildlife or wildlife product shipments that cleared customs (and thus should be considered legal trade), there is information on seizures within this dataset. Within the column “disposition”, values of “S” represent seized shipments. I’ve verified that there are thousands of data records for seized reptiles/reptile products that could be relevant to your study.

Page 2, Figure 1:

There seems to be something unusual happening with the presentation of data within this figure? Typically, the different portions of the Venn diagram represent distinct portions of the dataset, but that doesn’t seem to be the case here. For example, the text and figure would seem to indicate that the authors detected 2,754 reptile species in the online trade. However, detailed inspection of the figure seems to suggest that the shared online trade-LEMIS species (1,898), the shared online trade-CITES species (683), and the species present in all three datasets (622) is actually MORE (3,203) species than the online trade pool as a whole. This is the case in the data presentation for all three datasets. Perhaps the numbers reported as the shared portion across multiple datasets are not actually mutually exclusive data subsets (as I think they should be)?

Page 2, Figure 1:

The CITES listings given for the online traded species ($80 + 523 + 48$) don’t add up to the total declared CITES listed species (650) for that dataset.

Page 2, Figure 1:

You have the unenviable task of dealing with taxonomic issues across three different datasets here. Not trying to make your work more difficult, but I think it would be worth reviewing the unique reptile species names you’re using from LEMIS to make sure they are all in fact unique species. There could very easily be minor misspellings or synonymies present, which would mean that you are currently overestimating the unique reptile species present in that trade dataset. At the very least, I would give a qualifying statement about this, explaining the fact that these names, in their raw form, are not taxonomically standardized.

Page 2, paragraph 1:

It may be useful to include a brief (maybe just a sentence) explanation of the fact that you collected both a 2019 snapshot and a temporal sequence for online trade data. On my first read of the manuscript, this was a confusing part of the results, and I needed the methods to fully understand what exactly was being described here.

Page 2, “This contrasts to the species reported by CITES trade database, where 90% of species are listed in a CITES appendix (appendix-2: 81% 613/757; appendix-1 11% 94/844).”:

Why do these two figures not have the same denominator? Shouldn't the number of reptiles in the CITES dataset be consistent across the two calculations (844, I believe)? Also, why are Appendix III species not summarized in the same manner here?

Page 2:

It's probably going to be useful for your readers to mention why all species reported in the CITES data (844) are not actually CITES-listed (only 757 are). I presume this is because of country-specific monitoring and reporting of trade in the CITES Trade Database of species that are not actually on the CITES appendices?

Page 3, text and Figure 2C:

I'm rather confused about Figure 2C. First, it's showing the number of unique species that were never observed in another year? I think this requires a little more explanation in the figure caption, and I would change the y-axis label to reflect the fact that these are supposed to be unique species. Second, the average number of unique species per year values given in text seem suspicious given what the figure shows. It appears that the number of unique “all names” is substantially larger than the number of unique species name in each year, and the number of unique “all names” seems > 50 in at least 9 of the years shown. Yet the text states that the average unique “all names” per year is only 36.6 compared to 35.7 unique species names per year. Maybe I'm missing something, but the figure would suggest more drastic differences. Third, why is the CITES data plotted here, and is it also supposed to be the number of unique species by the same definition? It wouldn't seem there are on the order of 75 unique reptile species added to CITES each year, *especially* 75 species that do not appear in the CITES Trade Database at any point thereafter. So I'm unclear what this CITES data represents and why exactly it's relevant (doesn't seem to be referenced specifically anywhere in text?).

Page 4, “...represents 59% of trade events...”:

I think it's really important here and throughout to be explicit which dataset you're referring to when you reference trade or “trade events.” I presume this is only CITES trade events?

Page 4, “The commercial focus of CITES is further reflected in the regulation of fashion targeted species: 100% of crocodiles and 52% of Testudines, compared with 9% of lizards and 4% of snakes.”:

Do you mean to say that 100% of crocodiles are covered by CITES or that 100% of the crocodylian species covered by CITES are fashion-targeted species? How did you make that determination? This ambiguity in interpretation applies to all the taxa mentioned in this sentence.

Page 4, “In total 43.2% of CITES reptile trade events (and individuals) are from wild caught animals (44.4% from captivity or ranched)...”:

This is probably going to be very confusing for readers, so I think you want to be extremely clear which CITES source codes you used for your categorization of wild-caught animals. I agree ranched animals could belong in this category, but I'm not sure why captive-sourced animals (presumably source code C?) would be considered wild-caught (in fact, this would seem to be the exact opposite of their true status)?

Page 4, “Data from LEMIS shows that for 92% of species have wild-caught individuals imported, and only 44% of species had captive bred individuals imported.”:

Why not also report the number of LEMIS reptile trade records that are wild-caught versus captive-bred? It’s fine to include this particular metric, but it seems more unintuitive to report just the number of *species* for which there are *any* wild-caught or captive-bred transactions as opposed to the actual number of reptile trade records (or individuals) that are wild-caught versus captive-bred.

Page 5, Figure 3:

I would suggest making panel D into panel B and simply showing percentage of species in trade from all countries (rather than having an arbitrary species richness cutoff). Of course, your new plot would show that some places have a very high portion of their species in trade, even if they don’t have a large absolute number of species. But that’s what panel A is for. With data for all countries in panel A and panel B, readers can get the most comprehensive understanding of your data.

Page 6, Figure 4:

This is already very strong, but it might be worth verifying that the LEMIS species detections you’re reporting are in fact commercial trade. My thought was that some of the early species detections *could* be scientific trade that was documented by LEMIS (and hence maybe not as much of a conservation concern). I spot-checked one species (*Uroplatus giganteus*), which indeed seems to be involved in the commercial trade soon after its description. I think it would just make the figure all that more convincing if it was explicitly recording appearance in the *commercial* trade (all online trade is commercial, I assume).

Page 9, “We ceased cycling through search pages when a URL returned a 404 error, or when 100 pages had been cycled through. 100 pages were surveyed to prevent endless cycling back onto initial pages, or deriving errors from misinterpreting the number of search pages returned, whilst still exceeding the number of pages on most sites.”:

I can see the need for this limit, but could it have led to any bias in your search results towards species that appear earlier in the alphabet (i.e., 100 pages was not enough to characterize the complete stock list of a given site)? Is there any way to verify to yourself and your readers that you in fact pulled the complete species list for every site?

Page 11, “For examination of CITES coverage over time (species detected from Internet Archive pages) we used the more stringent single name matching because of the added complexity of a changing list of CITES species and the assumption that new CITES listings would use the most recently accepted name.”:

Where is the corresponding analysis in-text? The only obvious temporally-based CITES analysis is in Figure 1C, which doesn’t seem to have to do with the proportion of species in the online trade in a given year that were covered by CITES...

Page 11, “For LEMIS species counts we included those only listed to genus level, for example *Anolis* spp. would be counted as a species alongside *Anolis carolinensis* and *Anolis smaragdinus* etc.”:

I think it would be very important to mention how many of the distinct LEMIS reptile “species” you’re reporting are in fact these generic species declarations. You’re artificially inflating your number of traded species by including these in the count (even if we think LEMIS or any other legal wildlife trade database is actually a limited window onto the full scope of the wildlife trade).

Page 12, “Website names/URLs have been redacted to preserve their anonymity.”:

I’m not sure the justification for anonymization here? Certainly, I understand potential privacy issues, but all of the websites you scraped are presumably openly accessible to the public already. And having the complete website information seems relevant for any reader who wants to follow up on and vet the results of your study.

Minor Comments

Page 1:

“...unsustainability exploitation...” should be “...unsustainable exploitation...”

Page 1, “Although awareness of the scale of biodiversity loss is growing; assessments...”:
I think you need a comma rather than a semi-colon.

Page 1, “...potentially leaving thousands of traded species largely unmonitored...”:
It doesn't *potentially* leave them largely unmonitored. It *does* leave them largely unmonitored, correct?

Page 1, “At least 21 species have had their entire wild...”:
Incomplete sentence.

Page 1, “...how rapidly exploitation can impact new species.”:
I know it's a technicality, but maybe call these “newly described species”?

Page 1, “...System (LEMIS), of these only CITES trade...”:
The comma here should be a period starting a new sentence. Also, I think you mean to say, “...only the CITES trade...”

Page 3, “...discovered in our 2019 snapshot varied over time...”:
It would be helpful to explicitly mention the timeframe (2004-2018) here. Initially, the figure was a bit confusing because I was looking for the place where the online trade number equaled 834 species. But then I realized the figure timeline is only from 2004-2018.

Page 3, “...remaining comparatively since...”:
I think there's a word missing here?

Page 3, Figure 3 caption, “Trend in number of online trended species”:
Typo.

Page 5, “The true number of newly described species likely is much greater than 135...”:
The true number of newly described species *in trade* you mean?

Page 7, “But LEMIS data indicated that 91% of traded species include wild collected specimens.”:
Earlier in the text (page 4) this number was 92%?

Page 7, “The USA instigated the Lacey act to prevent animal trafficking in 1900 (24). The law not only recognised CITES listed species, but additionally included species with local regulations on trade and export.”:
This should be reworded. Right now, it almost seems like you are saying that the Lacey Act directly addressed CITES-listed species since its origination. Yet CITES didn't exist until the 1970s, as you state on page 1 of the manuscript.

Page 8, “For lower-value species banning trade from key-regions may not drive trade “underground” as can happen with higher value species (10).”:
Is reference 10 the appropriate reference for this statement?

Page 8, “...36% of reptile species are in trade; four times more species are than monitored via CITES.”:
Semi-colon should be a comma.

Page 10, “Overall, our species keyword list comprised of all scientific and common names from both Reptile Database and CITES (Data S2), with an average of 5.82 ± 0.06 s.e. per species and grand total of 64,342 terms (standard error (s.e.) calculated using the *pracma* package (48)).”:
Define “s.e.” the first time it is used in text.

Page 11, “We retrieved LEMIS data (v.1.1.0) via R using the *lemis* package (57).”:
I think that citing our *Scientific Data* Data Descriptor paper (<https://doi.org/10.1038/s41597-020-0354-5>) or the Zenodo repository (<https://doi.org/10.5281/zenodo.3565869>) would be better citations than the R package alone. (Readers should be able to accurately reference the paper and Zenodo repository into the foreseeable future even if the R package goes through updates.)

Page 12, “...where ignored for this analysis.”:
“where ignored” should be “were ignored”

Page 20, Figure S3:
Shouldn't your observed sample coverage value intersect the x-axis at the number of years for which you actually have data, which is 15 (2004-2018) rather than what's shown in the figure?

Pages 21-22, Figure S4:
I wonder if this would be more useful to readers simply as a table or series of tables? It's not as exciting as a visualization, but if people want the information that's represented in the country-specific bars/pie charts, that's currently very difficult to judge accurately from the visualization.

Reporting Summary, Research sample section:
As stated early, it may not be completely accurate to say that the subset of LEMIS data you analyzed represents legal trade (if some of the data are in fact from seized shipments).

Reporting Summary, Timing and spatial scale section, “The resulting sample covered web pages from 2012 to 2019.”:
I believe you mean 2002 to 2019?

Reporting Summary, Timing and spatial scale section, “LEMIS data covered a period from 2000 to 2019 and represents trade into the USA.”:
The LEMIS data you report using only contains data from 2000-2014.

Reviewer #2:

Remarks to the Author:

This study evaluated the scope of the global reptile trade using online databases and trade inventories. As the authors claimed in their results, the reptiles are arguably one of the most neglected taxa threatened by global trade. Through automated online data collection, the authors are able to document reptilian trade preferences, and regulation and monitoring gaps as currently observed. This is an important study and a much needed one for the reptilian conservation community and regulatory bodies. There are many valuable outcomes from this study that could really help us to protect the global reptiles threatened by trade. In fact, I do not have too many issues with the general approach, analyses, and the not-so-surprising findings.

However, the research questions and findings are not entirely novel, though the methodology may be considering the number of languages used and temporal extent of online data used. One of the main issues is that the authors omitted some key references in wildlife trade highly relevant to this study. It is hard to know why prominent references like Scheffers et al. 2019 and Frank and Wilcove 2019, both recently published in Science, were ignored. This is troubling since both studies are widely considered as groundbreaking work in the recent wildlife trade literature. In particular, Frank and Wilcove's work highlighted the similar issues facing threatened species. That is, there existed a lag time between trade and protection for the published commentary and between discovery and trade appearance for the new submission. If either of these papers is discussed, the reader would be less convinced about the novelty of this study.

Other minor issues included grammatical errors scattered throughout the manuscript. For example, see paragraph from line 29. Fig. 4C. There is no reason to color the bars using a series of colors when they don't mean much.

Reviewer #3:

Remarks to the Author:

The major claims of the paper are that over 36% of reptile species are exploited, and over three quarters of those are not covered by international trade regulation. This is useful information as it gives an idea (and some quantitative information) of the proportion of reptile species in trade that are covered by international regulation (ie CITES) and therefore helps provide wider context for other trade analyses, but of course this is only for reptiles, and is still likely to be an under-estimate. Another major finding is that nearly 4000 reptile species were found to be traded.

The paper also demonstrates that endangered or range-restricted species, with hotspots in Asia are traded, and that exploitation can occur soon after description, as has been shown for other species in trade such as orchids. These particularly threatened species should be highlighted for conservation action. The other major claim is that wild collection is widespread, potentially impacting 91% of species – but this is from the LEMIS and CITES data and the proportions of wild versus captive individuals in trade are not provided – just that a certain species has been traded from wild origin – so these data should be interpreted carefully. The concluding statement of the abstract suggests that a reversal of

the status quo is needed, requiring proof of sustainability before trade is permitted – this should influence thinking/stimulate discussion in this field, particularly in how such a move could be operationalised.

I do have some comments which I have outlined below.

A major comment relates to the lack of methodology regarding analysis of the CITES and LEMIS data. There is now a lot of literature surrounding use and misuse of the CITES database in particular, but the same should apply for the LEMIS database.

In Methods line 368 you link to the CITES Trade Dashboards for retrieval of data which I think is incorrect? (I am presuming you downloaded data from the actual CITES Trade Database, rather than the summary figures from the dashboards?)

It seems that you have mostly extracted species counts from these data rather than numbers, but in order for a researcher to reproduce this work, a note on these methodology need to be added.

For example what was the time frame for the data downloaded, how were they downloaded (gross reports/comparative reports etc), if comparative, did you use importer-reported or exporter-reported data sets?

In line 105-110 it appears some further analyses of the CITES data inform these results but there is no methodology on this. For example:

Line 106-107 – ‘Critically endangered species are primarily used commercially (94-96%)...’ how did you arrive at these percentages? (are these all sources, purposes, terms, units of trade etc?)

‘*Crocodylus siamensis* represents 50% of trade events’ how did you arrive at ‘trade events’ is this proportion of overall numbers? For what time period? For what sources/terms/units?

Line 113 – what do you mean by ‘trade events’? Did you download the new shipment level trade data? The data on the usual CITES trade database are not presented by individual trade events or shipments (see the trade database guide and references below)

Examples of some key sources on this:

Pavitt, A., Stafford, C., Tallwin, O., Vovk, E., Price, B., Banks, S., ... & Malsch, K. (2019). What is the reality of wildlife trade volume? Understanding CITES trade data—A response to Berec et al. *Biological Conservation*, 230, 195-196.

Robinson, J. E., & Sinovas, P. (2018). Challenges of analyzing the global trade in CITES-listed wildlife. *Conservation Biology*, 32(5), 1203-1206.

I also believe it should be made very clear that the online analyses is restricted to those species traded live for pets. Therefore the finding that over 36% of reptile species are (internationally) traded is even more likely to be an underestimate as other species may be traded online for food and products, and

these species may not be captured by the online search, CITES or the LEMIS database. This information informs interpretation of the overlap between different data sources.

There are some areas of text which suffer from over-cutting of text and additional words are needed for clarity.

Other comments:

Abstract

Line 4 – ‘unsustainable’ not ‘unsustainability’

Main text

Line 22 – it is not strictly true that ‘the regulations primarily protect large, commercially traded, charismatic species....’ Several large horticulturally important groups are listed on CITES and over 70% of CITES species are orchids. Consider rephrasing.

Line 36 – The final sentence is misleading given that the final reference relates to birds, not reptiles – be more specific here / clarify

Line 47 – you refer to ‘CITES trade portal’ – avoid ambiguity/be consistent in referring to CITES Trade Database

Fig 1 caption – this should read number of reptile species detected – as it currently reads it could suggest number of individual reptiles.

What time frame are these data from? Include this information

Be explicit in the figure that the online trade data includes those traded live for pets

Line 72 – I don’t understand this statement ‘This contrasts to the species reported by CITES trade database, where 90% of species are listed in a CITES appendix.....’ Is this because species are recorded in the CITES trade database that are not included on CITES appendices, do you include Appendix III here or are you restricting to Appendix I and II? - clarify

Line 80-82 – This first sentence needs a few words adding to explain you used web archive data, so it makes sense. I know this is explained in methodology but at this point all you have referred to is 2019 snapshot data.

Line 97 – ‘detected’ not ‘detecting’

Line 104 – link to figure 1 missing?

Line 119 – what do you mean by ‘diversity’?

Line 145 – Add 'in trade' to the following sentence 'The true number of newly described species in trade is likely much greater than...'

Discussion

Line 180 – consider adding 'and a minimum of 79% of traded species are not subject to CITES trade regulation... (given that online analyses only focused on pet trade).'

Line 185-189 – I think you need to acknowledge that wild trade is not necessarily bad for species and conservation in all cases – especially when part of regulated and monitored projects where local counterparts are receiving benefits from the trade and incentives are generated for conservation. Consider work by Dilys Roe and Rosie Cooney amongst others and the following paper which discusses the possible implications of wild versus ranched/captive reptiles in trade. I don't believe this needs a lot of focus but acknowledgment of complexities through addition of a sentence will allow it to come across more balanced.

Robinson, J. E., Griffiths, R. A., John, F. A. S., & Roberts, D. L. (2015). Dynamics of the global trade in live reptiles: Shifting trends in production and consequences for sustainability. *Biological Conservation*, 184, 42-50.

REVIEWER COMMENTS

Reviewer #1 (Remarks to the Author):

Please see attached my major and minor comments to the authors.

In this study, the authors have sought to characterize the legal wildlife trade in reptiles at a global scale. This is a worthy research goal, and the results are likely to be of broad interest to the conservation community. While the literature already contains various large-scale, primarily descriptive wildlife trade papers, I believe this manuscript distinguishes itself in a number of ways:

- 1) The authors have made a substantial effort to quantify the online trade in reptiles. This is a relatively novel, but critical, data source to consider, and the methods used for this portion of the study should be relevant to conservationists seeking to quantify the online wildlife trade across disparate taxa.
- 2) The authors have made the effort to integrate their novel online trade data with other notable existing wildlife trade data sources, namely the CITES Trade Database and the US-centric LEMIS dataset.
- 3) This manuscript addresses the short timeframe over which newly described reptile species may be impacted by the global wildlife trade. I think the authors are right to emphasize the importance of this particular result; this is a specific consequence of the wildlife trade that is rarely addressed in other studies, and the implications for rapid overexploitation of newly described species are alarming.

For the reasons above, I think this manuscript is deserving of eventual publication. At the moment, however, there are a number of outstanding issues that I believe are in need of improvement. Primarily, these are related to communication and interpretation of the study results. I have organized my specific feedback into major and minor comments (see below). I do not think any of my suggestions should require substantial data re-analysis, but addressing these issues will make for a more easily interpretable paper. Finally, I would note that there were relatively minor but somewhat pervasive grammatical issues (i.e., simple typos, problems with sentence construction, etc.) that should be addressed prior to publication. Some of these I've highlighted in my minor comments, but that is not a comprehensive list.

Response: Thank you, we hope this analysis can be used to inform better measures to inform conservation and ensure that trade is more sustainable, we feel that gaps in current approaches urgently need highlighting and hope that this research will better enable that. We have now gone through the text to make it easier to interpret and to fix issues stated to make it as useful as possible.

Major Comments

Reviewer: Page 1, "Piecemeal assessments fail to reveal...":

This seems a little vague. I think it's important to emphasize what you view as the limitations of prior work since that helps to distinguish your current efforts. So, in what particular ways do you

view these assessments as subpar? A focus on too few species? Inadequate data? A focus on limited geographic regions?

Response: We added the following to text

“Piecemeal assessments (focusing on a small subset of species, or locations based on variable methods”

Reviewer: Page 1, “...only CITES trade portal includes a percentage of seized specimens...”: This is not strictly true. While the vast majority of LEMIS data represents wildlife or wildlife product shipments that cleared customs (and thus should be considered legal trade), there is information on seizures within this dataset. Within the column “disposition”, values of “S” represent seized shipments. I’ve verified that there are thousands of data records for seized reptiles/reptile products that could be relevant to your study.

Response: Thank you, we have added a quantification of the number of entries that were classified as seized. This should provide sufficient context to support our statements that the bulk of trade recorded in LEMIS and CITES is legal.

“CITES and LEMIS trade databases includes a percentage of seized specimens (LEMIS, 14969/724656 2.07% of entries categorised as seized, CITES for snapshot 1384/49457 2.8%, overall 12775/328168 4% (actual number likely much lower once volumes accounted for due to large imports from commercial importers)), with all other trade legal and often international, and online trade at least purporting to be legal.”

Reviewer: Page 2, Figure 1:

There seems to be something unusual happening with the presentation of data within this figure? Typically, the different portions of the Venn diagram represent distinct portions of the dataset, but that doesn’t seem to be the case here. For example, the text and figure would seem to indicate that the authors detected 2,754 reptile species in the online trade. However, detailed inspection of the figure seems to suggest that the shared online trade-LEMIS species (1,898), the shared online trade-CITES species (683), and the species present in all three datasets (622) is actually MORE (3,203) species than the online trade pool as a whole. This is the case in the data presentation for all three datasets. Perhaps the numbers reported as the shared portion across multiple datasets are not actually mutually exclusive data subsets (as I think they should be)?

Response: We originally plotted the venn diagram with non-mutually exclusive divisions. We see how this could be misleading, so have revised which numbers are displayed in the sections. Now each species is only counted within one section of the venn diagram, and noted it in the legend to ensure this is clear.

Reviewer: Page 2, Figure 1:

The CITES listings given for the online traded species (80 + 523 + 48) don’t add up to the total declared CITES listed species (650) for that dataset.

Response: Corrected. We have double-checked the code, and this appears to be a typo.

Reviewer: Page 2, Figure 1:

You have the unenviable task of dealing with taxonomic issues across three different datasets here. Not trying to make your work more difficult, but I think it would be worth reviewing the unique reptile species names you’re using from LEMIS to make sure they are all in fact unique species. There could very easily be minor misspellings or synonymies present, which would mean that you are currently overestimating the unique reptile species present in that trade

dataset. At the very least, I would give a qualifying statement about this, explaining the fact that these names, in their raw form, are not taxonomically standardized.

Response: We used the same process for all datasets, online trade, CITES trade database and LEMIS, where we attempted to match them to the Reptile Database species list. In the case of LEMIS species names, only species that could be matched to a synonym present in Reptile Database were included in counts. We describe the impact of the synonymisation process on raw species counts in the supplementary text, but have added text to the methods to make it clear how we undertook this process on LEMIS data and the impacts of genus-only listings.

“As for CITES species lists, we matched the unstandardised LEMIS names to those present in Reptile Database (operating as our backbone nomenclature), leading to both synonymisations and splits. A LEMIS name was converted to a Reptile Database name if it matched any current, common, or historically used name. By LEMIS naming there were 639 instances of genus level listing, that were matched to 510 Reptile Database names. Of the 510 converted names, 442 appeared in other sources suggesting genus level listings in LEMIS did not inflate species counts. Those that failed to be converted were not included in total species counts, as final counts were entirely based on Reptile Database naming.”

Reviewer: Page 2, paragraph 1:

It may be useful to include a brief (maybe just a sentence) explanation of the fact that you collected both a 2019 snapshot and a temporal sequence for online trade data. On my first read of the manuscript, this was a confusing part of the results, and I needed the methods to fully understand what exactly was being described here.

Response: We added *“Analysis included both a “snapshot” of species currently present on reptile trading websites, and a longitudinal trend from the most “species rich” website using a “web-archive” to view both current availability and change over time.”*

Reviewer: Page 2, “This contrasts to the species reported by CITES trade database, where 90% of species are listed in a CITES appendix (appendix-2: 81% 613/757; appendix-1 11% 94/844).”: Why do these two figures not have the same denominator? Shouldn’t the number of reptiles in the CITES dataset be consistent across the two calculations (844, I believe)? Also, why are Appendix III species not summarized in the same manner here?

Response: Both those calculations should have been based on the 844 count of traded species, matching the calculations on online trade and LEMIS. We have updated and changed the values accordingly. We elected to report appendix 3 values only via figure 1, previous comments on the manuscript highlighted an overabundance of percentages that hindered reading. Avoiding reporting appendix 3 in text with percentage helped text flow, and the considerable lower counts in appendix 3 do not reveal much more than patterns evident in appendix 1 and 2 percentages.

“where 90% of species are listed in trade are in a CITES appendix (e.g appendix-2: 72.6% 613/844; appendix-1 11% 94/844).”

Reviewer: Page 2:

It’s probably going to be useful for your readers to mention why all species reported in the CITES data (844) are not actually CITES-listed (only 757 are). I presume this is because of country-specific monitoring and reporting of trade in the CITES Trade Database of species that are not actually on the CITES appendices?

Response: As of April 2020 897 species of reptile have CITES appendices, but those in trade in the portal include both listed and unlisted species, the following text has been added

“Not all CITES trade database species are covered by CITES appendices. Species reported in the CITES database may come from seizures (2.8-4%), or from shipments which include both species with CITES appendices, and unlisted species.”

Reviewer: Page 3, text and Figure 2C:

I’m rather confused about Figure 2C. First, it’s showing the number of unique species that were never observed in another year? I think this requires a little more explanation in the figure caption, and I would change the y-axis label to reflect the fact that these are supposed to be unique species. Second, the average number of unique species per year values given in text seem suspicious given what the figure shows. It appears that the number of unique “all names” is substantially larger than the number of unique species name in each year, and the number of unique “all names” seems > 50 in at least 9 of the years shown. Yet the text states that the average unique “all names” per year is only 36.6 compared to 35.7 unique species names per year. Maybe I’m missing something, but the figure would suggest more drastic differences. Third, why is the CITES data plotted here, and is it also supposed to be the number of unique species by the same definition? It wouldn’t seem there are on the order of 75 unique reptile species added to CITES each year, *especially* 75 species that do not appear in the CITES Trade Database at any point thereafter. So I’m unclear what this CITES data represents and why exactly it’s relevant (doesn’t seem to be referenced specifically anywhere in text?).

Response: We have changed the way the data is displayed in 2C. The area plot was misrepresenting the data, stacking the *all name* and *sci. name* counts, the new line plot avoids this issue. We double-checked our calculation of the mean, and it appears the discrepancy was entirely in data display of stacked counts. Further we have added a second y axis to clarify CITES data line is displaying a count of CITES protected species traded, not connected to the unique species. We have updated and expanded the figure caption to reflect this. We have also added a reference in the text to this trend.

“Dashed line shows the raw count of CITES listed species annually (right y-axis), determined by exact matching of Reptile Database name with CITES listed name.” & “Raw counts of species indicate the consistent presence of CITES protected species in the trade (75.5 ±5.81; Fig. 2C)”

Reviewer: Page 4, “...represents 59% of trade events...”:

I think it’s really important here and throughout to be explicit which dataset you’re referring to when you reference trade or “trade events.” I presume this is only CITES trade events?

Response: Thanks-added

“represents 59% of trade listings in CITES (i.e recorded trade statistics between different countries with different sources, types etc”

Reviewer: Page 4, “The commercial focus of CITES is further reflected in the regulation of fashion targeted species: 100% of crocodiles and 52% of Testudines, compared with 9% of lizards and 4% of snakes.”:

Response: added *“have CITES appendix listings, which coincides with those used commercially in fashion”*

Reviewer: Do you mean to say that 100% of crocodiles are covered by CITES or that 100% of the crocodylian species covered by CITES are fashion-targeted species? How did you make that determination? This ambiguity in interpretation applies to all the taxa mentioned in this sentence.

Page 4, “In total 43.2% of CITES reptile trade events (and individuals) are from wild caught animals (44.4% from captivity or ranched)...”:

Response: Added “,with for example 20 of the 22 species of crocodylian listed with commercial or personal “purposes””, most species included the term “leather, garments, cloth or sometimes carving” within the database and the purpose of “commercial” or “personal” use

“Other trends (i.e. percentage of species coming from different sources or with different statuses were calculated in excel using basic approaches to quantify listings with different qualities (i.e. seized, wild, commercial, personal use etc) and quantify the percentage with that status within CITES etc. For more extensive analysis of multiple factors summary statistics were used in ArcMap after joining fields to connect species data from traded specimens of the three data sources with Red List assessments. This provided some simple statistics to further understand patterns as detailed in text.”

Reviewer: This is probably going to be very confusing for readers, so I think you want to be extremely clear which CITES source codes you used for your categorization of wild-caught animals. I agree ranched animals could belong in this category, but I’m not sure why captive-sourced animals (presumably source code C?) would be considered wild-caught (in fact, this would seem to be the exact opposite of their true status)?

Response: Rephrased and clarified-sorry for confusion

“In total 43.2% of CITES reptile trade events (and individuals) are from wild caught animals (whereas 44.4% are from captivity or ranched (with some of these ranched individuals probably also collected from the wild as eggs or young)).”

Reviewer: Page 4, “Data from LEMIS shows that for 92% of species have wild-caught individuals imported, and only 44% of species had captive bred individuals imported.”:

Why not also report the number of LEMIS reptile trade records that are wild-caught versus captive-bred? It’s fine to include this particular metric, but it seems more unintuitive to report just the number of *species* for which there are *any* wild-caught or captive-bred transactions as opposed to the actual number of reptile trade records (or individuals) that are wild-caught versus captive-bred.

Response: We have expanded on this section, using LEMIS data to provide more details on the extent of wild-capture.

“In terms of trade-events, LEMIS lists 63.0% (63.2% excluding seized shipments) as wild-sourced (456,722/724,655), nearly twice that listed as originating from captive, ranching, or commercial breeding (35.6% 258,021/724,655). LEMIS data suggest that at least 58.2% of traded individuals are taken from the wild (61,390,757/105,536,941), compared to 41.3% from breeding activities (105,536,941/43,611,039). Examination of RedList status (for species that could be connected to a Red List status) reveal that 5.17% (99/1914) of LEMIS traded species are Critically Endangered, and a further 7.11% are Endangered (136/1914), illustrating the likely overlap with wild-capture and vulnerability.”

Reviewer: Page 5, Figure 3:

I would suggest making panel D into panel B and simply showing percentage of species in trade from all countries (rather than having an arbitrary species richness cutoff). Of course, your new plot would show that some places have a very high portion of their species in trade, even if they don’t have a large absolute number of species. But that’s what panel A is for. With data for all countries in panel A and panel B, readers can get the most comprehensive understanding of your data.

Response: We have moved D to B, (and B to C, C to D) as suggested. We feel that giving both percent and numbers helps understand the impacts more clearly. Showing percentage everywhere makes depauperate species look highly exploited, whereas that percentage is most significant when it remains high when there are a large number of species, but we have changed the order as recommended and added a supplemental figure as suggested here (Figure S5) which includes percentage collected even in depauperate areas so that readers can understand the impact on both diverse and more species poor regions.

Reviewer: Page 6, Figure 4:

This is already very strong, but it might be worth verifying that the LEMIS species detections you're reporting are in fact commercial trade. My thought was that some of the early species detections *could* be scientific trade that was documented by LEMIS (and hence maybe not as much of a conservation concern). I spot-checked one species (*Uroplatus giganteus*), which indeed seems to be involved in the commercial trade soon after its description. I think it would just make the figure all that more convincing if it was explicitly recording appearance in the *commercial* trade (all online trade is commercial, I assume).

Response: Thank you for highlighting this aspect, it's an important point to make. We have added details on the balance between commercial versus noncommercial trade in relation to the overall numbers reported. For the description to trade figure we filtered out the non-commercial species revising down numbers and updated figure 4 accordingly. In the case of the lag between description and trade, this filtering of non-commercial species modified the number of species we could detect in a particular year; therefore, the mean and SE of the lag time slightly.

“Overlap between online trade results and the species reported by LEMIS corroborate the online search results; although, 2.18% of trade events listed in LEMIS are for non-commercial purposes, and 634 species are only traded non-commercially (775 species using Reptile Database standard). Of the 775 non-commercial species, only 459 species do not appear in CITES or online trade lists..”

Methods regarding figure 4 - “We excluded species listed as only being traded for LEMIS non-commercial purposes”

“Ninety-two of these species could be connected (from the 2000-2018 timeframe) to a year of first appearance (Fig. 4A), while the other 41 species were only detected in the 2019 snapshot data so their initial date of appearance in the trade is unknown (Fig. 4B). The true number of newly described species in trade likely is much greater than 133, as splits of species complexes are likely being traded under older names.”

Reviewer: Page 9, “We ceased cycling through search pages when a URL returned a 404 error, or when 100 pages had been cycled through. 100 pages were surveyed to prevent endless cycling back onto initial pages, or deriving errors from misinterpreting the number of search pages returned, whilst still exceeding the number of pages on most sites.”:

I can see the need for this limit, but could it have led to any bias in your search results towards species that appear earlier in the alphabet (i.e., 100 pages was not enough to characterize the complete stock list of a given site)? Is there any way to verify to yourself and your readers that you in fact pulled the complete species list for every site?

Response: It is extremely difficult to verify we have collected the entire stock list for every website. The diversity of web site set-ups made comprehensive searches difficult, hence the hierarchy of search methods we needed to employ. However, the point of systematic bias remains important to address, we have reviewed a random subsample of the “cycle” searched

web pages to check the ordering of species. In 4/10 sites we could not determine how species lists had been ordered, in 6 they were ordered by date, and in 1 it was ordered by popularity. We believe the combination and inconsistency of ordering would not lead to a systematic bias towards species near the beginning of the alphabet, but recognise that ordering may have led to greater variation between websites. We have added details to the methods to help clarify this variation:

“We performed a post-hoc review of ten sites searched using a “cycle” search method to check whether species ordering could have led systematic biases for names near the beginning of the alphabet or price. For four websites we could not determine how species were ordered, for six websites species listings were ordered by date, and for one website species were ordered by popularity, thus even for sites with more pages we feel the results will not be impacted by biases given the inconsistency of approach for order of entries on different sites.”

Reviewer: Page 11, “For examination of CITES coverage over time (species detected from Internet Archive pages) we used the more stringent single name matching because of the added complexity of a changing list of CITES species and the assumption that new CITES listings would use the most recently accepted name.”:

Where is the corresponding analysis in-text? The only obvious temporally-based CITES analysis is in Figure 1C, which doesn’t seem to have to do with the proportion of species in the online trade in a given year that were covered by CITES...

Response: We have now added a statement alongside the in-text description of 2C detailing what the CITES per year findings, along with a mean CITES species per year. This change is paired with changes to Fig.2C to make what is being displayed clearer. See response to comments on Page 3, text and Fig. 2C above concerning how 2C displays the count of species that are covered by CITES appendices.

Reviewer: Page 11, “For LEMIS species counts we included those only listed to genus level, for example *Anolis* spp. would be counted as a species alongside *Anolis carolinensis* and *Anolis smaragdinus* etc.”:

I think it would be very important to mention how many of the distinct LEMIS reptile “species” you’re reporting are in fact these generic species declarations. You’re artificially inflating your number of traded species by including these in the count (even if we think LEMIS or any other legal wildlife trade database is actually a limited window onto the full scope of the wildlife trade).

Response: We use the same name synonymisation technique on all datasets, converting the raw names provided by the databases to the Reptile Database standard, so a generic name would not be counted as specific species after cleaning of data, only in a stage of analysis. Values supplied in the supplementary text show how the overall number of species names change due to this before and after. But we have added numbers to better describe the impact of only sp. listings in LEMIS in the methods, providing an estimate of their prevalence in the overall species count. See response to comment Page 2, Figure 1. The fact that species can be imported under a generic name is also an indicator of the need from systemic change to enable exact quantitative monitoring.

Reviewer: Page12, “Website names/URLs have been redacted to preserve their anonymity.”: I’m not sure the justification for anonymization here? Certainly, I understand potential privacy issues, but all of the websites you scraped are presumably openly accessible to the public

already. And having the complete website information seems relevant for any reader who wants to follow up on and vet the results of your study.

Response: We leave it to the journal to decide based on their policy, we feel there may be legal implications and it may contravene some of the data sharing and privacy guidelines, so will follow the journals recommendation on how to proceed and if such data should be included.

As many websites are for classified advertisements and frequently update, exact reproduction of our dataset may be difficult. We have supplied all code we used to generate the dataset in the hope that, despite frequent changes to reptile selling websites, others can corroborate the broad patterns we show (namely the scope of species traded) even without the identical search engine results and reptile selling websites. We have also supplied materials to aid future running of the code: a template of the input website data to aid the web scraping on updated or new websites (Data S1), and the compiled keyword list (Data S2).

Minor Comments

Reviewer: Page1:

“...unsustainability exploitation...” should be “...unsustainable exploitation...”

Response: Corrected

Reviewer: Page1, “Although awareness of the scale of biodiversity loss is growing; assessments...”:

I think you need a comma rather than a semi-colon.

Response: Corrected

Reviewer: Page1, “...potentially leaving thousands of traded species largely unmonitored...”:

It doesn't *potentially* leave them largely unmonitored. It *does* leave them largely unmonitored, correct?

Response: Corrected

Reviewer: Page1, “At least 21 species have had their entire wild...”:

Incomplete sentence.

Response: edited “*At least 21 species have had their entire wild populations harvested by collectors using species descriptions (5), and numerous other populations have suffered declines due to over-collection*”

Reviewer: Page1, “...how rapidly exploitation can impact new species.”:

I know it's a technicality, but maybe call these “newly described species”?

Response: Corrected

Reviewer: Page1, “...System (LEMIS), of these only CITES trade...”:

The comma here should be a period starting a new sentence. Also, I think you mean to say, “...only the CITES trade...”

Response: Corrected

Reviewer: Page3, “...discovered in our 2019 snapshot varied over time...”:

It would be helpful to explicitly mention the timeframe (2004-2018) here. Initially, the figure was a bit confusing because I was looking for the place where the online trade number equaled

834 species. But then I realized the figure timeline is only from 2004-2018.

Response: added “*compared to the changes that occurred in trade between 2004-2018*”

Reviewer: Page3, “...remaining comparatively since...”:

I think there’s a word missing here?

Response: added “*consistent*”

Reviewer: Page3, Figure 3 caption, “Trend in number of online trended species”:

Typo.

Response: Corrected “*traded*”

Reviewer: Page5, “The true number of newly described species likely is much greater than 135...”:

The true number of newly described species *in trade* you mean?

Response: Corrected, added “*in trade*”

Reviewer: Page7, “But LEMIS data indicated that 91% of traded species include wild collected specimens.”:

Earlier in the text (page 4) this number was 92%?

Response: Corrected, rounding error.

Reviewer: Page7, “The USA instigated the Lacey act to prevent animal trafficking in 1900 (24). The law not only recognised CITES listed species, but additionally included species with local regulations on trade and export.”:

This should be reworded. Right now, it almost seems like you are saying that the Lacey Act directly addressed CITES-listed species since its origination. Yet CITES didn’t exist until the 1970s, as you state on page 1 of the manuscript.

Response: reworded “*While the Lacey act preceded CITES, it expanded to recognise CITES listed species, in addition to species with local regulations on trade and export.* ”

Reviewer: Page8, “For lower-value species banning trade from key-regions may not drive trade “underground” as can happen with higher value species (10).”:

Is reference 10 the appropriate reference for this statement?

Response: Thanks this should have been reference 11 (Courchamp et al., 2006), we have now double checked numbers at the end (this is a legacy of former edits and changes in some of the references)

Reviewer: Page8, “...36% of reptile species are in trade; four times more species are than monitored via CITES.”:

Semi-colon should be a comma.

Response: corrected

Reviewer: Page10, “Overall, our species keyword list comprised of all scientific and common names from both Reptile Database and CITES (Data S2), with an average of 5.82 ± 0.06 s.e. per species and grand total of 64,342 terms (standard error (s.e.) calculated using the pracma package (48)).”:

Define “s.e.” the first time it is used in text.

Response: added

Reviewer: Page11, “We retrieved LEMIS data (v.1.1.0) via R using the lemis package (57).”: I think that citing our *Scientific Data* Data Descriptor paper (<https://doi.org/10.1038/s41597-020-0354-5>) or the Zenodo repository (<https://doi.org/10.5281/zenodo.3565869>) would be better citations than the R package alone. (Readers should be able to accurately reference the paper and Zenodo repository into the foreseeable future even if the R package goes through updates.)

Response: We have added in both citations alongside the citation for the LEMIS package.

Reviewer: Page12, “...where ignored for this analysis.”:
“where ignored” should be “were ignored”

Response: Corrected

Reviewer: Page20, Figure S3:

Shouldn't your observed sample coverage value intersect the x-axis at the number of years for which you actually have data, which is 15 (2004-2018) rather than what's shown in the figure?

Response: For the sample assessments we included the full 17 years 2002-2019 as we based the total species count off all years and all detections. We only excluded 2002, 2003 and 2019 from the regression analysis because of the low page count. We have made small changes to the methods text to make this clearer. This has also highlighted an error in the annotation in Fig. 4 that suggests we relied on data from LEMIS exclusively between 2000-2003, we have now corrected that to say to 2000-2001. We have also added the exclusion of these years to the main text where temporal trends are reported.

“(we excluded 2002, 2003 and 2019 because of a lack of archived pages)”

Reviewer: Pages 21-22, Figure S4:

I wonder if this would be more useful to readers simply as a table or series of tables? It's not as exciting as a visualization, but if people want the information that's represented in the countryspecific bars/pie charts, that's currently very difficult to judge accurately from the visualization.

Response: Some patterns, like the disparity between African and Asian country regional patterns would be hard to discern from a table, so we find visuals more intuitive, but we have added a Data S6 as a complement which provides a full table of all data used.

Reviewer: Reporting Summary, Research sample section:

As stated early, it may not be completely accurate to say that the subset of LEMIS data you analyzed represents legal trade (if some of the data are in fact from seized shipments).

Reporting Summary, Timing and spatial scale section, “The resulting sample covered web pages from 2012 to 2019.”:

I believe you mean 2002 to 2019?

Reporting Summary, Timing and spatial scale section, “LEMIS data covered a period from 2000 to 2019 and represents trade into the USA.”:

The LEMIS data you report using only contains data from 2000-2014.

Response: We could not find where in text where exactly this refers to but have been through the draft to check all years referenced in text are now correct.

Reviewer #2 (Remarks to the Author):

This study evaluated the scope of the global reptile trade using online databases and trade inventories. As the authors claimed in their results, the reptiles are arguably one of the most neglected taxa threatened by global trade. Through automated online data collection, the authors are able to document reptilian trade preferences, and regulation and monitoring gaps as currently observed. This is an important study and a much needed one for the reptilian conservation community and regulatory bodies. There are many valuable outcomes from this study that could really help us to protect the global reptiles threatened by trade. In fact, I do not have too many issues with the general approach, analyses, and the not-so-surprising findings.

However, the research questions and findings are not entirely novel, though the methodology may be considering the number of languages used and temporal extent of online data used. One of the main issues is that the authors omitted some key references in wildlife trade highly relevant to this study. It is hard to know why prominent references like Scheffers et al. 2019 and Frank and Wilcove 2019, both recently published in Science, were ignored. This is troubling since both studies are widely considered as groundbreaking work in the recent wildlife trade literature. In particular, Frank and Wilcove's work highlighted the similar issues facing threatened species. That is, there existed a lag time between trade and protection for the published commentary and between discovery and trade appearance for the new submission. If either of these papers is discussed, the reader would be less convinced about the novelty of this study.

Response: Thankfully the editors and other reviewers recognise the novelty and importance in our analyses, but we thank you for providing an additional reference.

The Frank and Wilcove paper is a useful and relevant addition to the paper and has now been added (added the following to the discussion “Studies have demonstrated that CITES is consistently behind the IUCN in species assessment and inclusion (28). Until population impacts are known and assessments complete, trade-bans from specific regions, where exports include threatened species, should also be considered. ”. However, due to former discussions with colleagues working for the IUCN, TRAFFIC and other NGOs, Scheffers' paper is widely thought to be a misinterpretation of CITES data, see <https://news.mongabay.com/2019/10/misuse-of-wildlife-trade-data-jeopardizes-efforts-to-protect-species-and-combat-trafficking-commentary/>. Thus we felt no need to reference it in this paper, because of concerns that the measures of “what is traded” are mixed with “what is measured”. The desire to discover which species are traded outside the knowledge of large bodies like the IUCN and CITES was partly the motivation for our online trade searches. This is especially important for reptiles that face greater gaps in assessments than some other vertebrate groups.

Reviewer: Other minor issues included grammatical errors scattered throughout the manuscript. For example, see paragraph from line 29. Fig. 4C. There is no reason to color the bars using a series of colors when they don't mean much.

Response: We have now edited and proofed the draft, and removed the colour from Figure 4C- thanks for comments.

Reviewer #3 (Remarks to the Author):

The major claims of the paper are that over 36% of reptile species are exploited, and over three quarters of those are not covered by international trade regulation. This is useful information as it gives an idea (and some quantitative information) of the proportion of reptile species in trade that are covered by international regulation (ie CITES) and therefore helps provide wider context for other trade analyses, but of course this is only for reptiles, and is still likely to be an under-estimate.

Another major finding is that nearly 4000 reptile species were found to be traded.

The paper also demonstrates that endangered or range-restricted species, with hotspots in Asia are traded, and that exploitation can occur soon after description, as has been shown for other species in trade such as orchids. These particularly threatened species should be highlighted for conservation action. The other major claim is that wild collection is widespread, potentially impacting 91% of species – but this is from the LEMIS and CITES data and the proportions of wild versus captive individuals in trade are not provided – just that a certain species has been traded from wild origin – so these data should be interpreted carefully. The concluding statement of the abstract suggests that a reversal of the status quo is needed, requiring proof of sustainability before trade is permitted – this should influence thinking/stimulate discussion in this field, particularly in how such a move could be operationalised.

I do have some comments which I have outlined below.

Reviewer: A major comment relates to the lack of methodology regarding analysis of the CITES and LEMIS data. There is now a lot of literature surrounding use and misuse of the CITES database in particular, but the same should apply for the LEMIS database.

Response: Thanks, we agree (especially with regards to CITES) some “high impact” publications have done this in the past and the results were very misleading. Here we have tried to use best practice throughout, and avoided quantifying the different units within several of our datasets because of both issues with reporting reliably and consistency and because of the different units used. Thanks for highlighting the issues here, we have addressed each point, and made amendments in text to better explain approaches used. We have added more detail on the CITES and LEMIS analysis complete in the methods. Further the code supplied alongside the manuscript provides a supplementary description of the exact ways LEMIS data was summarised.

Reviewer:In Methods line 368 you link to the CITES Trade Dashboards for retrieval of data which I think is incorrect? (I am presuming you downloaded data from the actual CITES Trade Database, rather than the summary figures from the dashboards?)

Response: Thanks, this has been corrected - <https://trade.cites.org>

Reviewer:It seems that you have mostly extracted species counts from these data rather than numbers, but in order for a researcher to reproduce this work, a note on these methodology need to be added.

Response: We retrieved counts of text keyword hits from websites, but we did not use counts in the analysis for a number of reasons (mainly because they do not reflect the number of individuals): species may be re-advertised by the same seller, and sellers may advertise multiple individuals with one advertisement. We felt that they were not accurate enough to really indicate abundance of each species, this has now been detailed in methods. We have added the following text. The supplied code provides a supplementary description of how we summarised the keyword search data.

“Though our online search analysis provided the number of mentions per species per page, we do not detail these numbers because sellers may list multiple individuals at once, sellers may post the same advertisement numerous times, or that advertisements can be repeated on different pages within the same website. Therefore, numbers derived from online analysis did not provide a reliable estimate of numbers per species for sale, and we elected to restrict analysis to binary species appearances.”

Reviewer: For example what was the time frame for the data downloaded, how were they downloaded (gross reports/comparative reports etc), if comparative, did you use importer-reported or exporter-reported data sets?

Response: CITES data was downloaded on the dates and for the periods shown in the methods, for the snapshot and time-series respectively

Reviewer: In line 105-110 it appears some further analyses of the CITES data inform these results but there is no methodology on this. For example:

Line 106-107 – ‘Critically endangered species are primarily used commercially (94-96%)...’ how did you arrive at these percentages? (are these all sources, purposes, terms, units of trade etc?)

Response: These are based on listings within the CITES database, quantities were not used due to the combination of units and material types, thus for analysis referring to quantities from CITES it only refers to the number of trade listings and not the absolute quantities due to misreporting of numbers and varied units. Text has now been added to clarify this point, and additional text has been added in the methods to ensure methods used are clear. *“Compared to online trade and LEMIS, there were fewer species belonging to each RedList category in CITES trade database apart from Critically Endangered (59 species total). Critically Endangered species are predominantly used commercially (94-96% of trade listings), and dominated by a small number of species (e.g. *Crocodylus siamensis* represents 59% of trade listings (i.e. recorded trade statistics between different countries with different sources, types etc) and *Eretmochelys imbricata* a further 14.5%) over a CITES 2016-2019 snapshot. The commercial focus of CITES is further reflected in the regulation of fashion targeted species: 100% of crocodiles and 52% of Testudines, compared with 9% of lizards and 4% of snakes have CITES appendix listings, which coincides with those used commercially in fashion, with for example 20 of the 22 species of crocodylian listed with commercial or personal “purposes”.*

We have also added details to the methods:

“Other trends (i.e. percentage of species coming from different sources or with different statuses were calculated in excel using basic approaches to quantify listings with different qualities (i.e. seized, wild, commercial, personal use etc) and quantify the percentage with that status within CITES etc. For more extensive analysis of multiple factors summary statistics were used in ArcMap after joining fields to

connect species data from traded specimens of the three data sources with RedList assessments. This provided some simple statistics to further understand patterns as detailed in text.”

Reviewer: ‘Crocodylus siamensis represents 50% of trade events’ how did you arrive at ‘trade events’ is this proportion of overall numbers? For what time period? For what sources/terms/units?

Response: Trade events has been changed to trade listings and refers to identified trade within the CITES database “(e.g. *Crocodylus siamensis* represents 59% of trade listings (i.e recorded trade statistics between different countries with different sources, types etc) and *Eretmochelys imbricata* a further 14.5%) over a CITES 2004-2019 snapshot”

Reviewer: Line 113 – what do you mean by ‘trade events’? Did you download the new shipment level trade data? The data on the usual CITES trade database are not presented by individual trade events or shipments (see the trade database guide and references below)

Examples of some key sources on this:

Pavitt, A., Stafford, C., Tallowin, O., Vovk, E., Price, B., Banks, S., ... & Malsch, K. (2019). What is the reality of wildlife trade volume? Understanding CITES trade data—A response to Berec et al. *Biological Conservation*, 230, 195-196.

Robinson, J. E., & Sinovas, P. (2018). Challenges of analyzing the global trade in CITES-listed wildlife. *Conservation Biology*, 32(5), 1203-1206.

Response: Thanks, added “(i.e recorded trade statistics between different countries with different sources, types etc”. Unlike LEMIS and online we cannot quantify the exact volumes traded in CITES due to variation in units of what is recorded (though we can quantify the approximate numbers of exchanges) and can use this to measure complexity of supply chains and level of demand rather than a measure of quantity.

Similarly the temporal trend issue brought up in the Robinson & Sinovaa 2018 paper is important, but we do not base our conclusions on examinations of temporal trends. Temporal assessments are largely restricted to the online trade and LEMIS data, while also flawed, we could better correct for sampling biases (page count regression). We also did not use CITES data in our assessments of lag from description to trade.

The ambiguity regarding purpose codes is also worth addressing as we do report those. In this case, the conflation between “commercial” (covering both industry uses and commercial pet trade) and “personal” (more likely to be largely pet trade) would impact the exact values we’d report. We do not see a way of correcting for this, but feel that the main purpose of the manuscript is not undermined –highlighting the worrying extent of species traded with little oversight. We also feel the statement that CITES is more commercially focused than the other sources would change, as evidenced by findings that side-step reported purposes (e.g. “*The commercial focus of CITES is further reflected in the regulation of fashion targeted species: 100% of crocodiles and 52% of Testudines, compared with 9% of lizards and 4% of snakes have CITES appendix listings*”).

We appreciate having these sources brought to our attention, and hope that given our avoidance of using quantities in CITES which they highlight as a major issue that this will not impact on our results, though

are glad that these issues are being highlighted more broadly as they have real implications for our understanding of quantity of trade, rather than as we have explored number of species.

I also believe it should be made very clear that the online analyses is restricted to those species traded live for pets. Therefore the finding that over 36% of reptile species are (internationally) traded is even more likely to be an underestimate as other species may be traded online for food and products, and these species may not be captured by the online search, CITES or the LEMIS database. This information informs interpretation of the overlap between different data sources.

Response: We added “largely for pets” as the other databases also include a small number for fashion, even if most are for the pet trade. We mention in text that others are used for food or medicine, but they are likely to be impossible to fully assess, we also highlight this on our supplemental section on caveats. We also added further information in the discussion as follows

“CITES aims to ensure that wildlife trade is sustainable, yet it largely focuses on only the most economically valuable species traded in large volumes, leaving species which may have niche markets, are lesser-known, or range-limited, unprotected and vulnerable to trade. Key differences exist between CITES listed species being traded under CITES monitoring, and those for sale online or in LEMIS. For CITES data, the majority comprises a small number of species, traded in high volumes for the fashion trade; whereas those online were almost exclusively for the pet trade (though there is also a huge medicinal market including over 284 reptile species which was not explored here (16))”

Reviewer: There are some areas of text which suffer from over-cutting of text and additional words are needed for clarity.

Other comments:

Reviewer: Abstract

Line 4 – ‘unsustainable’ not ‘unsustainability’

Response: corrected

Reviewer: Main text

Line 22 – it is not strictly true that ‘the regulations primarily protect large, commercially traded, charismatic species....’ Several large horticulturally important groups are listed on CITES and over 70% of CITES species are orchids. Consider rephrasing.

Response: We added the word “animals” as though orchids are very charismatic for plants, many plants are neglected in regulations

Reviewer: Line 36 – The final sentence is misleading given that the final reference relates to birds, not reptiles – be more specific here / clarify

Response. Text has been edited to read “and numerous other populations have suffered declines due to over-collection”

The reference has also been changed to Schlaepfer MA, Hoover C, Dodd CK. Challenges in evaluating the impact of the trade in amphibians and reptiles on wild populations. *BioScience*. 2005 Mar 1;55(3):256-64. and a second reference added

Todd BD, Willson JD, Gibbons JW. The global status of reptiles and causes of their decline. *Ecotoxicology of amphibians and reptiles*. 2010 Jun 2;47:67.

Reviewer: Line 47 – you refer to ‘CITES trade portal’ – avoid ambiguity/be consistent in referring to CITES Trade Database

Response: Done, thanks

Reviewer: Fig 1 caption – this should read number of reptile species detected – as it currently reads it could suggest number of individual reptiles.

Response: corrected

Reviewer: What time frame are these data from? Include this information

Be explicit in the figure that the online trade data includes those traded live for pets

Response: The data included both the 2019 snapshot and the longitudinal assessment shown on figure 2 from 2000-2018 (to allow for complete years). We have added this to the figure legend and expanded the text to make this clearer.

Reviewer: Line 72 – I don’t understand this statement ‘This contrasts to the species reported by CITES trade database, where 90% of species are listed in a CITES appendix.....’ Is this because species are recorded in the CITES trade database that are not included on CITES appendices, do you include Appendix III here or are you restricting to Appendix I and II? - clarify

Response: We added the following “*Other species listed in the CITES database may either come from seizures, or from shipments which included both species with CITES appendices, and unlisted species.*”

Reviewer: Line 80-82 – This first sentence needs a few words adding to explain you used web archive data, so it makes sense. I know this is explained in methodology but at this point all you have referred to is 2019 snapshot data.

Response: We added “*Analysis included both a “snapshot” of species currently in trade based on analysis of 151 websites, and in addition a longitudinal trend from the most “species rich” website based on a “web-archive” to view both current availability and change over time.*”

Reviewer: Line 97 – ‘detected’ not ‘detecting’

Response: corrected

Reviewer: Line 104 – link to figure 1 missing?

Response: added

Reviewer: Line 119 – what do you mean by ‘diversity’?

Response: added “species” to clarify

Reviewer: Line 145 – Add ‘in trade’ to the following sentence ‘The true number of newly described species in trade is likely much greater than...’

Response: added

Reviewer: Discussion

Line 180 – consider adding ‘and a minimum of 79% of traded species are not subject to CITES trade regulation... (given that online analyses only focused on pet trade).

Response: added

Reviewer: Line 185-189 – I think you need to acknowledge that wild trade is not necessarily bad for species and conservation in all cases – especially when part of regulated and monitored projects where local counterparts are receiving benefits from the trade and incentives are generated for conservation. Consider work by Dilys Roe and Rosie Cooney amongst others and the following paper which discusses the possible implications of wild versus ranched/captive reptiles in trade. I don’t believe this needs a lot of focus but acknowledgment of complexities through addition of a sentence will allow it to come across more balanced.

Robinson, J. E., Griffiths, R. A., John, F. A. S., & Roberts, D. L. (2015). Dynamics of the global trade in live reptiles: Shifting trends in production and consequences for sustainability. *Biological Conservation*, 184, 42-50.

Response: Thanks, we have added the suggested reference and the additional text “*Thus though studies have found that captive breeding and ranching can provide alternative livelihoods and thus enable conservation, for this to be achieved mechanisms to prevent laundering are needed, and the cost of rearing animals cannot substantially exceed that for collecting from the wild*”

Reviewers' Comments:

Reviewer #3:

Remarks to the Author:

While some aspects of the paper have been improved, based on the clarifications received I have major concerns regarding the methodology, analysis and interpretation of the CITES data.

Firstly my comment regarding adequate methodology for the CITES data has not been adequately addressed I can see the correct download url/link to the database has now been added for the CITES data, but no other information provided. For example, there is no mention of the report type used (comparative, gross/net trade report), and what was requested in the download (dates/all reptiles?).

I strongly recommend the authors look at other papers which have analysed data from the CITES Trade Database and include a section on this in the methods.

Secondly, the authors have clarified in the methods and elsewhere that they have used "number of listings rather than absolute quantities because of inaccuracies in reporting in different units". This is very unclear but I think they mean that they have summarized the data based on the number of rows which appear in the data – which they previously referred to as 'trade events', and they have now changed the wording to 'number of listings'. This is a fundamental problem.

The individual rows in the CITES data do not refer to trade events – they are data amalgamated for the year which match on import and export country, and all sources, purposes, terms and units. So for example all Python regius from Ghana to the US, traded live, from wild source, for one year (say 2015) will be summarized on that line. The references explaining this were provided in the previous review.

(This is presuming the authors downloaded comparative tabulations – which is what is presumed but is also not clear as they have not reported this. It also assumes the authors have not downloaded the newly available shipment level database – but again – not clear)

So when the authors report xx% of listings were from commercial/wild source etc, these data are not meaningful. The line I just described may contain 1200 individuals, whereas the next line reporting captive bred python regius from the US to Germany may contain 4 individuals for that year. The next line may be referring to 10,000 watch straps made from a monitor species.

The methods referring to the online analysis are very comprehensive and the results based on the online searches are sound and interesting. I get the impression the CITES and LEMIS data may have been a later addition to this paper. As a result the methodology, analyses and interpretation of the CITES data are inadequate. The same may issues may apply to the LEMIS data but I have less experience with this data set.

I recommend that the authors either make a thorough analysis of these data to the same level of attention that has been applied to the online data, or they remove all quantification of these data and ONLY use the species lists/identification of species in trade, from these data sources. This would involve the removal of some paragraphs in the results but not a fundamental restructure in my opinion.

I also still find some sections of the results and writing clunky and unclear. I have outlined a few extra comments below.

Main text, 2nd para, 2nd sentence – This now reads as if CITES only protects large charismatic mammals which is clearly not true.

Main text, 2nd para, last sentence - “Piecemeal assessments...” This sentence sounds a bit derogatory as there are several comprehensive global and/or regional assessments of reptile trade.

Results, scale of trade, third para - This paragraph is clunky – It seems obvious that the CITES data will contain a high number of CITES listed species!

Also what do you mean in the first line of this paragraph that ‘overlap between online trade results and species reported by LEMIS corroborate online search results’? Not clear

Results, the threat from trade, second para – this paragraph is problematic for reasons explained above. What you mean here, based on my understanding, is that ‘50% of the rows in amalgamated data referred to *Crocodylus siamensis*’, and so on, which is not meaningful information.

“Compared to online trade and LEMIS, there were fewer species belonging to each RedList category in CITES trade database apart from Critically Endangered (59 species total). Critically Endangered species are predominantly used commercially (94-96% of trade listings) [what if the other 5-6% of rows or ‘trade listings’ refers to several millions of reptiles traded for scientific purposes, for example – although not likely this is to illustrate why these data are not meaningful] , and dominated by a small number of species (e.g. *Crocodylus siamensis* represents 59% of trade listings (i.e recorded trade statistics between different countries with different sources, types etc – [this addition does not clarify]) and *Eretmochelys imbricata* a further 14.5%) over a CITES 2016-2019 snapshot. The commercial focus of CITES is further reflected in the regulation of fashion targeted species: 100% of crocodiles and 52% of Testudines, compared with 9% of lizards and 4% of snakes have CITES appendix listings, which coincides with those used commercially in fashion, with for example 20 of the 22 species of crocodylian listed with commercial or personal “purposes” [what is the relevance/significance of this last point?].

Results, origin of traded species, 1st para – this whole paragraph is not meaningful for reasons described above. The addition of the word ‘substantial’ in this paragraph is misplaced. 2.5% and 8.8% does not represent ‘substantial?’

Results, origin of traded species, 2nd para – I do not think these data are meaningful and they could be misunderstood. For example, “92% of species have wild caught individuals imported”. But without any quantification of these data many of these species may have had an isolated scientific specimen, or single blood sample, traded from wild where nearly 99% is from captive source.

Results, origin of traded species, 3rd para first sentence – Fig 3A seems to refer to the number of traded species not species diversity.

Fig 3 - you need to make clear in legend this is just using the online trade

REVIEWER COMMENTS

Reviewer #3 (Remarks to the Author):

While some aspects of the paper have been improved, based on the clarifications received I have major concerns regarding the methodology, analysis and interpretation of the CITES data.

Firstly my comment regarding adequate methodology for the CITES data has not been adequately addressed I can see the correct download url/link to the database has now been added for the CITES data, but no other information provided. For example, there is no mention of the report type used (comparative, gross/net trade report), and what was requested in the download (dates/all reptiles?).

Response: We have updated the methods section, it now reads: *CITES data was retrieved from <https://trade.cites.org/#> on 2020-05-13) using the comparative tabulations for all “reptilia” and the appropriate years (the snapshot of 2019, and 2004-2019) to download all reptile species exported over this time.*

I strongly recommend the authors look at other papers which have analysed data from the CITES Trade Database and include a section on this in the methods.

Response: Most papers on CITES data focus on the volumes exported whereas we largely focus on what is being traded and the implications. Though we cite some papers that focus on the CITES data (i.e. Frank & Wilcove 2019), but have now added more details to methods, and provided a much more in depth breakdown of the CITES data.

“Though the research focused on the percentages of species vulnerable to trade based on various forms of IUCN and CITES category we made some efforts to quantify the proportions of items with different statuses within CITES and LEMIS. Quantifications were made using a number of different approaches. Online assessments were not directly quantified due to the possibility of listing the same individuals multiple times, or having mixed batches of specimens with variable numbers. For CITES we used the summary statistics tool in ARCMAP to quantify the means and totals for the numbers exported and imported, and the range for each species or endangerment status is provided in text (or a single number if they were the same). Redlist status was associated with the data by joining the scientific name field between the two databases. Sums were made for various sources, purposes and endangerment statuses for CITES data using this same approach, based on the 2004-2019 data from the CITES trade portal. “Terms” (i.e. skins) were also explored, recategorising the standard terms (57 were used for reptiles) into nine (i.e. fashion), then summing the total item number imported and exported and determining the percentage.

Secondly, the authors have clarified in the methods and elsewhere that they have used “number of listings rather than absolute quantities because of inaccuracies in reporting in different units”. This is very unclear

but I think they mean that they have summarized the data based on the number of rows which appear in the data – which they previously referred to as ‘trade events’, and they have now changed the wording to ‘number of listings’. This is a fundamental problem.

Response: The CITES data has now been quoted with the percentages of items exported and imported for each designation examined, the range is given where the two differ and a single value is given when an integer value was the same. This is not a perfect comparison given the issue with units, but is at least highly indicative of trends. To give an idea of complexity in trade for some species we also give a brief analysis of some of the most widely traded species and the numbers of countries involved in the trade of such species.

The individual rows in the CITES data do not refer to trade events – they are data amalgamated for the year which match on import and export country, and all sources, purposes, terms and units. So for example all Python regius from Ghana to the US, traded live, from wild source, for one year (say 2015) will be summarized on that line. The references explaining this were provided in the previous review.

Response: See above, this has now been changed. This is only a very minor component of the paper, as we want to examine how many species are impacted, rather than the degree of impact on those species (which because of units etc is harder to gauge), and actually there was a very good alignment between the number of “trade-event” and the number of items imported and exported in the same categories in terms of percentage, but we now have the more easily comprehensible metrics. As an alternative we now use the sum totals imported and exported for any given designation and provide the range. This element is mainly provided to demonstrate that in reptiles CITES largely applies to species targeted in huge volumes for fashion trade, and that species traded for other purposes, such as pets are rarely included and consequently vulnerable.

(This is the presuming the authors downloaded comparative tabulations – which is what is presumed but is also not clear as they have not reported this. It also assumes the authors have not downloaded the newly available shipment level database – but again – not clear)

Response: Yes comparative tabulations were used, this has now been detailed in the methods. The shipment level data appears not to be available online, but the standard CITES one meets the needs of this study given that the main fields would likely be the same as data would be formatted in a standard way.

“using the comparative tabulations for all “reptilia” and the appropriate years (the snapshot of 2019, and 2004-2019) to download all reptile species exported over this time.”

So when the authors report xx% of listings were from commercial/wild source etc, these data are not meaningful. The line I just described may contain 1200 individuals, whereas the next line reporting captive bred python regius from the US to Germany may contain 4 individuals for that year. The next line may be referring to 10,000 watch straps made from a monitor species.

Response: Close alignment existed between the “trade events” and volumes exported and imported, but we have now changed the figures to provide information based on numbers exported and imported and determined percentages based on these numbers

The methods referring to the online analysis are very comprehensive and the results based on the online searches are sound and interesting. I get the impression the CITES and LEMIS data may have been a later addition to this paper. As a result the methodology, analyses and interpretation of the CITES data are inadequate. The same may issues may apply to the LEMIS data but I have less experience with this data set.

Response: The LEMIS data is somewhat different, especially as the individual sales (and companies) are listed. We have added a sentence to the methods clarifying the structure of LEMIS data. “LEMIS data includes shipment level records of imports to the USA, alongside information pertaining to purpose, quantity, origin, date among other metadata.”

Furthermore as the first reviewer created the major repository of LEMIS data and found no faults in our treatment of data we can be confident that this data was treated and analysed appropriately

I recommend that the authors either make a thorough analysis of these data to the same level of attention that has been applied to the online data, or they remove all quantification of these data and ONLY use the species lists/identification of species in trade, from these data sources. This would involve the removal of some paragraphs in the results but not a fundamental restructure in my opinion.

Response: The species list results are comparable, species detected in the online search, LEMIS data, and CITES trade database underwent the same procedures as online data (once the webscraping was complete) –filtering, synonymisation, then summary. We have also added the following to the methods to clarify that all data sources underwent the same process and are compared at that point. “Final species counts are based on unique Reptile Database names, after this synonymization/split process.”

“Following the online webscraping the same types of analysis and cleaning were applied to all three databases.”

Reliably extracting stock levels or number of individuals from online trade was not possible, but because quantity is an important aspect to consider for species with small distribution or ranges, we felt it necessary to include the best available number concerning this. The inclusion of quantities for only CITES and LEMIS, despite the flaws in datasets, does not undermine the primary message of the article, namely that thousands of species are being traded, and few are subject to international regulation.

I also still find some sections of the results and writing clunky and unclear. I have outlined a few extra comments below.

Response: Thanks, the article has been proofed for comprehensibility and flow

Main text, 2nd para, 2nd sentence – This now reads as if CITES only protects large charismatic mammals which is clearly not true.

Response: The text now states “The regulations primarily protect commercially traded, “charismatic” animal species; only recently covering lesser-known species (e.g. pangolins-2016).” we removed “large” from the assertion

Main text, 2nd para, last sentence - “Piecemeal assessments...” This sentence sounds a bit derogatory as there are several comprehensive global and/or regional assessments of reptile trade.

Response: Existing assessments for reptiles only use the CITES trade database (i.e. reference 24), or as stated are regional, or taxa specific. As we show here these assessments are not comprehensive, and better cataloging (i.e. based on the standards utilised by LEMIS) are needed even for a clear understanding of legal trade. We have also rephrased the sentence: “Assessments focusing on a small subset of species or locations (often using variable methods), can fail to reveal the true extent of the wildlife trade and the impact on traded species, especially within groups such as reptiles”

Results, scale of trade, third para - This paragraph is clunky – It seems obvious that the CITES data will contain a high number of CITES listed species!

Response: We included specific references to the number of non-CITES species in the CITES trade database on the behest of another reviewer. We feel it necessary to keep this section to make sure we/or readers are not making incorrect assumptions about the comprehensiveness of CITES (e.g. assuming all CITES species are traded and vice versa). Recent comments on published studies have highlighted the problems with that assumption (Kolby, 2019. Commentary in response to Scheffers et al., 2019. <https://news.mongabay.com/2019/10/misuse-of-wildlife-trade-data-jeopardizes-efforts-to-protect-species-and-combat-trafficking-commentary/>)

Also what do you mean in the first line of this paragraph that ‘overlap between online trade results and species reported by LEMIS corroborate online search results’? Not clear

Response: The sentence aimed to highlight the overlap in species between the two datasets, supporting the more experimental online searches with the (presumably) more mature customs data. We have simplified this sentence to highlight the overlap. “Overlap between online trade results and species reported by LEMIS was considerable (1,898 species)”

Results, the threat from trade, second para – this paragraph is problematic for reasons explained above. What you mean here, based on my understanding, is that ‘50% of the rows in amalgamated data referred to *Crocodylus siamensis*’, and so on, which is not meaningful information.

Response: The CITES statistics have now been provided inline with the reviewers recommendations based on exports and imports.

“Compared to online trade and LEMIS, there were fewer species belonging to each RedList category in CITES trade database apart from Critically Endangered (59 species total). Critically Endangered species are predominantly used commercially (94-96% of trade listings) [what if the other 5-6% of rows or ‘trade listings’ refers to several millions of reptiles traded for scientific purposes, for example – although not likely this is to illustrate why these data are not meaningful] , and dominated by a small number of species (e.g. *Crocodylus siamensis* represents 59% of trade listings (i.e recorded trade statistics between different countries with different sources, types etc – [this addition does not clarify]) and *Eretmochelys imbricata* a further 14.5%) over a CITES 2016-2019 snapshot. The commercial focus of CITES is further reflected in

the regulation of fashion targeted species: 100% of crocodiles and 52% of Testudines, compared with 9% of lizards and 4% of snakes have

Response: The CITES statistics have now been provided inline with the reviewers recommendations based on exports and imports.

CITES appendix listings, which coincides with those used commercially in fashion, with for example 20 of the 22 species of crocodilian listed with commercial or personal “purposes” [what is the relevance/significance of this last point?].

Response: That CITES is targeting species in the fashion trade, and neglects the pet trade, and largely scientific research. We feel it important to highlight how CITES functions and what is being targeted, and thus characterising the contents of the three different trade monitoring databases is important. This has now been made more evident.

“The purpose is largely listed as commercial (95-96%%), with a minority for personal use (0.1%), highlighting that within CITES monitored data the majority is for commercial (largely fashion) purposes, and neglects trade for other purposes such as the pet-trade.”

Results, origin of traded species, 1st para – this whole paragraph is not meaningful for reasons described above. The addition of the word ‘substantial’ in this paragraph is misplaced. 2.5% and 8.8% does not represent ‘substantial?’

Response: The CITES statistics have now been provided in-line with the reviewers recommendations based on exports and imports.

Results, origin of traded species, 2nd para – I do not think these data are meaningful and they could be misunderstood. For example, “92% of species have wild caught individuals imported”. But without any quantification of these data many of these species may have had an isolated scientific specimen, or single blood sample, traded from the wild where nearly 99% is from captive source.

Response: While it is true that the import could be a single scientific specimen, we highlighted earlier in the paper that only 2.18% of species in LEMIS are for non-commercial purposes meaning that wild collection is still impacting the vast majority of species, and that given the lack of regulation the true scope of this is difficult to assess. We further support the scale of wild collection with the number of traded individuals (44%). Granted, trade is likely targeting some species over others leading to a skew in the 92% and 44%, but without a priori assessments of trade impacts on populations the best trade data would be insufficient in determining what’s damaging to a particular species. Therefore, we feel reporting broad statistics is meaningful in the sense of highlighting how much work is required to comprehensively understand the trade’s impact on wild populations, and the number of species subjected to at least some collection of trade from the wild.

Results, origin of traded species, 3rd para first sentence – Fig 3A seems to refer to the number of traded species not species diversity.

Response: We have added “traded” to the sentence to make that clearer.

Fig 3 - you need to make clear in legend this is just using the online trade

Response: We have added “all” to the figure legend title, as it is from all species traded.

Thanks again for the opportunity, I hope we now meet all recommended edits!

Reviewers' Comments:

Reviewer #1:

Remarks to the Author:

This is my second review of the present manuscript. I still believe this paper contains important data (particularly the analysis of the online reptile trade), and there has been improvement in some areas, such as the Figure presentation. However, I also still have significant concerns about the clarity and accuracy of the data reported throughout, which prevent me from endorsing publication. The issues I have with the manuscript are very similar to those expressed by Reviewer #3, and while the authors have attempted to respond to our comments through two rounds of review, some of the core problems remain.

My read of the tension is this: the authors are caught between a high-level, species-level analysis and a more detailed characterization of the reptile trade. I think the authors are well-justified in limiting their description of the online reptile trade to just the species that occur in that trade; quantification of the online reptile trade may be too difficult or inaccurate given the methods employed. The analysis of the online reptile trade is, to me, the most novel part of the study and could stand by itself. However, the authors have attempted to bolster their work with the inclusion of both CITES and LEMIS trade data (an understandable aim), and this is where the problems enter in. If the paper is built around the synthesis of three disparate data sources, then it will only be successful to the degree that all three data sources are accurately analyzed and well-integrated. The CITES and LEMIS data *could* be used exclusively to document additional reptile species in the global reptile trade, which would be analogous to the type of information provided by the online analysis. But when the authors want to make claims about the sources of animals/products in the reptile trade, for example, this will require much more detailed scrutiny and description of the CITES and LEMIS data. It's not very meaningful to highlight that X% of species have *any* individuals being wild-caught (as multiple reviewers have pointed out). Rather, you must clearly summarize the data at an animal- or item-level to make a convincing case. I'm concerned that these sorts of analyses are still lacking in the present manuscript. For example, the LEMIS data appear to be summarized using rows of reptile data (rather than an analysis of counts of animals or items). Wildlife trade databases are extremely difficult to understand: when you say "item" we need to be sure you actually do mean numbers of items and which types of wildlife items you're referring to. Other elements of the manuscript and response to reviewers sow further confusion: in their second response to comments, the authors state that, "only 2.18% of species in LEMIS are for non-commercial purposes meaning that wild collection is still impacting the vast majority of species." Yet this statement conflates wildlife trade's potential purposes (i.e., non-commercial trade) and sources (i.e., wild vs. captive breeding).

At this point, I would ask the authors to carefully consider whether detailed descriptions of CITES and LEMIS data are vital to their message. I understand there may be frustration since these CITES and LEMIS analysis elements are unlikely to change the overall picture that many reptile species are present in a largely unregulated global trade. But if these analyses are included, I think they have to be accurate and communicated effectively. Alternatively, it could be that the paper is better served by cutting down some of the CITES and LEMIS material and focusing on the broader, species-level overview (as Reviewer #3 recommended in the last round of comments).

Please see my specific major and minor comments below.

Evan A. Eskew

Major Comments

Page 1, “The regulations primarily protect commercially traded, “charismatic” animal species; only recently covering lesser-known species (e.g. pangolins-2016).”

I know this sentence has been subject to previous comment, but it still reads as inaccurate. It now seems as if you mean to suggest CITES *only* protects charismatic animal species, which is clearly untrue given that the majority of listed species are plants.

Page 2, Figure 1

I previously commented that it would be worth reviewing the unique reptile species names you're using from LEMIS. There could very easily be minor misspellings or synonymies present. Your previous response partially addressed this issue by clarifying that taxonomies from all datasets were referenced against the Reptile Database species list. This addresses the concern that you could be double (or more) counting synonymous nomenclature for a single species. But it doesn't entirely resolve the concern. If you haven't thoroughly reviewed the unique LEMIS species names you're analyzing, it's possible that misspellings are resulting in undercounting (i.e., a species name in the database is a clear misspelling of a valid species name that doesn't get matched against the Reptile Database and is therefore left out of your analysis). At the very least, I would give a qualifying statement about this, explaining the fact that these names, in their raw form, are not taxonomically standardized. Ideally, you review the LEMIS taxonomy to verify why any names are not matching the Reptile Database and resolve ambiguities. But at the very least, your readers should be clearly told this is an outstanding issue in the analysis.

Page 4, Figure 2C

I still find this figure panel unnecessarily confusing. Why not simply separate the unique species observed in online trade over time and the CITES species present in online trade over time? These are completely distinct metrics with very different interpretations (one represents species that are not documented in any other year, the other does not). For this reason, I think the two sets of information would work better as two separate panels. At the very least, the y-axis labels should be more informative. Something like “Number of unique species in the online trade” and “CITES species present in the online trade”, respectively.

Page 4, Figure 2C

How could it be the case that a count of all names is ever less than strictly scientific names (as in the year 2007 here)? I think the distinction between these two categories deserves better explanation either here or in the Methods on page 13.

Page 5, “In terms of traded items, LEMIS lists 63.0% (63.2% excluding seized shipments) as wild-sourced (456,722/724,655), nearly twice that listed as originating from captive, ranching, and commercial breeding (35.6% 258,021/724,655).”

If I understand correctly, you've just filtered the LEMIS data for class Reptilia, leaving 724,655 rows of data, which you treat as “items.” To me this is a major issue, and one that is directly analogous to Reviewer #3's concerns about how you've treated CITES trade data in various iterations of the manuscript. Those rows of data you're analyzing are not items. Rather they could be representative of multiple reptile-derived products. This distinction matters in the sense that it affects every trade metric you report for LEMIS or CITES data that does not rely on a

gross, species-level analysis. To do these calculations accurately, they should actually be on the level of items, which will require more filtering of the data or a concentration on particular wildlife product descriptions (in the LEMIS data) or Terms (in the CITES data). For example, you might report the wild versus captive sourcing of only live animals or some specific reptile product like leather.

Page 8, Figure 4

I believe in the caption it should read, “The period 2000-2001 is only covered by LEMIS data.” I would also very, very explicitly mention that the LEMIS data has been filtered for commercial trade here (if indeed it has?). It’s not enough to bury this important information in the Methods.

Page 9, “...this totaled to over 63% of all imports.”

What exactly are you summarizing here? Wildlife product shipments? A sum of wildlife product items?

Page 10, “For lower-value species banning trade from key-regions may not drive trade “underground” as can happen with higher value species (11). Conservationists actively hindered trade ban implementation for birds on such grounds (29), but when eventually applied within Europe global bird trade decreased by 90% (30).”

This would seem to be a very generous read of the Reino et al. paper. First, that paper documents a decrease in bird trade when the EU implemented a trade ban, but it also notes that global trade flows were reshaped by the ban. Second, you claim that reptiles are “lower-value” species, yet your paper mentions small-ranged, endemic species and the Courchamp et al. paper you also cite explicitly lays out how rarity might be inversely related to value. So I’m not sure it’s obvious that rare reptiles *would* be low value in a trade ban scenario. Finally, and most importantly, you’re relying on the perceived effectiveness of trade bans as suggested by the Reino et al. paper, yet that manuscript uses CITES trade data, which you claim throughout this manuscript is a rather limited perspective on wildlife trade. And it doesn’t address the potential for redirection to illegal trade channels, which is one of the main concerns with trade bans. All this is to say, I think you need a more nuanced discussion of potential policy solutions for global reptile trade that doesn’t exclusively push for potentially ineffective or harmful trade bans.

Page 11, “We ceased cycling through search pages when a URL returned a 404 error, or when 100 pages had been cycled through. 100 pages were surveyed to prevent endless cycling back onto initial pages, or deriving errors from misinterpreting the number of search pages returned, whilst still exceeding the number of pages on most sites.” Through your revisions you’ve convinced me that stopping at 100 pages has not led to any alphabetical bias in your coverage of online reptile trade (since sites list reptiles in different schema). But you still haven’t assuaged the concern that you’ve not completely characterized the stock list of every site. This doesn’t need to be redone necessarily, but I would explicitly state in the methods that your online data scraping strategy may result in undercounts of species in online trade given that there are reptile trade sites you may have incompletely assessed.

Page 13, “We counted the unique species in two ways: using all names to detect species, and only scientific names.”

As mentioned in my comment on Figure 2C, I think the distinction/definitions here need much more precise explanation so that readers can better interpret your results.

Page 14, “For LEMIS species counts we included those only listed to genus level, for example *Anolis sp.* would be counted as a species alongside *Anolis carolinensis* and *Anolis smaragdinus* etc. As for CITES species lists, the unstandardised LEMIS names were matched to those present in Reptile Database (operating as our backbone nomenclature), leading to both synonymisations and splits. A LEMIS name was converted to a Reptile Database name if it matched any current, common, or historically used name. By LEMIS naming there were 639 instances of genus level listing, that were matched to 510 Reptile Database names. Of the 510 converted names, 442 appeared in other sources, suggesting genus level listings in LEMIS did not inflate species counts. Those that failed to be converted were not included in total species counts; as final counts were entirely based on Reptile Database explicit species naming. Final species counts from all data sources are based on unique Reptile Database names, after this synonymization/split process.”

I would be very specific here in the final sentence and state that your final species counts do not include any generic identifiers (if indeed they don't). The discussion here of the number of genus-level listings in LEMIS is almost a distraction from the issue because I thought, from your response to reviewers, that genus-level IDs don't make it into the species counts (shown in Figure 1) at all. But the phrase here “suggesting genus level listings in LEMIS did not inflate species counts” seems to state that the genus-level IDs in LEMIS matched with other genus-level IDs in other data sources (that were included in the final species counts)?

Page 14, “For CITES we used the summary statistics tool in ArxMap to quantify the means and totals for the numbers exported and imported...”

Yet earlier on this page you only mention gathering CITES export data? I think this is important because this ambiguity links in with many of the concerns Reviewer #3 has had throughout the review process: with the CITES data, it's very important that you be clear which dataset you're working with and how you are reporting trade statistics (i.e., importer- or exporter-reported data).

Minor Comments

Page 1, “Here we highlight the scope of the global reptile trade: over 36% of reptile species are traded largely for pets, over three quarters of which are not covered by international trade regulation, including numerous endangered or range-restricted species, with hotspots in Asia.”

This is a bit of a sprawling, hard-to-follow sentence. Given that it contains some of your paper's key results, I would consider revising for simplicity.

Page 2, “Only CITES and LEMIS trade databases include a percentage of seized specimens (LEMIS, 14969/724656 2.07% of individuals were categorised as seized, CITES 0.2-0.4% overall (imported 781421/197819654; exported 425156/206360017)).”

I would consider revising the punctuation in this sentence for clarity.

Page 3, “...2.18% of individuals listed in LEMIS are for non-commercial purposes...”

Is this meant to imply live individuals?

Page 3, “This contrasts to the species reported by CITES trade database, where 90% of species are listed in trade are in a CITES appendix...”

One too many “are”s here.

Page 3, “Not all CITES trade database species are covered by CITES appendices. Species reported in the CITES database may come from seizures (2.8-4%), or from shipments which include both species with CITES appendices, and unlisted species.”

I don't think this is a completely accurate description of the way these data arise. It's not simply the case that unlisted species are randomly caught up in the CITES reporting process because of their co-occurrence with CITES-listed species. Rather, there are other mechanisms for non-CITES-listed species to be reported in this database, including the EU Wildlife Trade Regulations (https://ec.europa.eu/environment/cites/species_en.htm).

Page 3, “Listings in LEMIS covered large volume imports to fashion companies (which are listed), as well as much smaller numbers of a diverse selection of species to other buyers.”

What does the “which are listed” phrase refer to? Listed on what?

Page 3, “Raw counts of species indicate the consistent presence of CITES protected species in the trade (75.5 ± 5.81 ; Fig. 2C).”

In which trade? I believe it's critical to point out this is online trade, correct?

Page 5, “In addition all 13 species imported into over 125 countries are crocodiles, pythons and monitors, and the three species exported from 125 countries or more include two crocodiles and one python species.”

This is all a bit difficult to follow. I assume you're talking about the 13 species in the CITES trade database that happen to fall within the five genera you previously mentioned?

Page 5, “The commercial focus of CITES is further reflected in the regulation of fashion targeted species: 100% of crocodiles and 52% of Testudines have a CITES appendix, compared with 9% of lizards and 4% of snakes have CITES appendix listings, which coincides with those used commercially in fashion, with 20 of the 22 species of crocodilian listed with commercial or personal “purposes”.”

This remains unclear to me. Do you mean to say that 100% of all crocodiles and 52% of all Testudines species globally are covered by CITES? Are the 22 crocodilian species only those listed by CITES or are you saying that there are 22 crocodilian species globally? Are the 20 species *exclusively listed* with commercial or personal purposes or those that were *ever listed* with commercial or personal purposes?

Page 5, “In total 53% of CITES reptile items traded are from wild caught animals (whereas 33-36% are from captivity).”

I think for the CITES data, it's always critical to mention whether you're discussing importer- or exporter-reported data.

Page 5, “The purpose is largely listed as commercial (95-96%%), with a minority for personal use (0.1%), highlighting that within CITES monitored data the majority is for commercial (largely fashion) purposes, and neglects trade for other purposes such as the pet-trade.”

This phrasing almost makes it sound like CITES is making an active decision to neglect pet trade. I would consider saying that pet trade is not “widely represented” in the CITES trade database or something similar.

Page 5, “Data from LEMIS shows that for 92% of species have wild-caught individuals imported...”

Don’t need the “for”

Page 5, “...has more than 50% of their reptile species covered by CITES regulations...”

Do you mean their *traded* reptile species?

Page 9, “...those for sale online or in LEMIS.”

This could be read ambiguously. Species aren’t for sale in LEMIS. Their trade is documented by LEMIS.

Page 9

I believe “Lacey act” should appear as “Lacey Act” here and throughout the manuscript.

Page 11, “...to retrieve a list of reptiles selling websites...”

Should be “reptile selling websites”

Page 12, “Five CITES listed species had no matching counterpart in the Reptile Database, we determined this was caused by minor spelling mistakes.”

Comma splice.

Page 23, Figure S3

Your response to my previous concerns about this Figure was a bit confusing. My comment led you to revise text in other parts of the manuscript, but the problem with this Figure remains: your caption states the data considered is from 2004-2018 (15 years) whereas the sample coverage value does not intersect the x-axis at year 15 but rather at some greater year value (I can’t tell which exactly). The response to reviewers was also confusing because it suggested that the years 2002-2019 would constitute 17 years of data when it’s actually 18.

Reporting Summary, “Research sample” section

It may not be completely accurate to say that the subset of LEMIS data you analyzed represents legal trade (if some of the data are in fact from seized shipments).

Reviewer #3:

Remarks to the Author:

The manuscript has been much improved given the adjustment to the CITES data and its presentation, and added clarity to the methods.

There are a few places where the authors refer to "individuals" which seems to suggest whole animals. Given that these data refer to a range of units (live, parts etc) i suggest, for consistency and clarity, that the authors use "items" as they have in most other occasions.

The manuscript now makes a compelling read and i wish the authors well with its publication.

Reviewer #1

This is my second review of the present manuscript. I still believe this paper contains important data (particularly the analysis of the online reptile trade), and there has been improvement in some areas, such as the Figure presentation. However, I also still have significant concerns about the clarity and accuracy of the data reported throughout, which prevent me from endorsing publication. The issues I have with the manuscript are very similar to those expressed by Reviewer #3, and while the authors have attempted to respond to our comments through two rounds of review, some of the core problems remain.

My read of the tension is this: the authors are caught between a high-level, species-level analysis and a more detailed characterization of the reptile trade. I think the authors are well-justified in limiting their description of the online reptile trade to just the species that occur in that trade; quantification of the online reptile trade may be too difficult or inaccurate given the methods employed. The analysis of the online reptile trade is, to me, the most novel part of the study and could stand by itself. However, the authors have attempted to bolster their work with the inclusion of both CITES and LEMIS trade data (an understandable aim), and this is where the problems enter in. If the paper is built around the synthesis of three disparate data sources, then it will only be successful to the degree that all three data sources are accurately analyzed and well integrated.

The CITES and LEMIS data *could* be used exclusively to document additional reptile species in the global reptile trade, which would be analogous to the type of information provided by the online analysis. But when the authors want to make claims about the sources of animals/products in the reptile trade, for example, this will require much more detailed scrutiny and description of the CITES and LEMIS data. It's not very meaningful to highlight that X% of species have *any* individuals being wild-caught (as multiple reviewers have pointed out). Rather, you must clearly summarize the data at an animal- or item-level to make a convincing case. I'm concerned that these sorts of analyses are still lacking in the present manuscript. For example, the LEMIS data appear to be summarized using rows of reptile data (rather than an analysis of counts of animals or items). Wildlife trade databases are extremely difficult to understand: when you say "item" we need to be sure you actually do mean numbers of items and which types of wildlife items you're referring to. Other elements of the manuscript and response to reviewers sow further confusion: in their second response to comments, the authors state that, "only 2.18% of species in LEMIS are for non-commercial purposes meaning that wild collection is still impacting the vast majority of species." Yet this statement conflates wildlife trade's potential purposes (i.e., non-commercial trade) and sources (i.e., wild vs. captive breeding).

At this point, I would ask the authors to carefully consider whether detailed descriptions of CITES and LEMIS data are vital to their message. I understand there may be frustration since these CITES and LEMIS analysis elements are unlikely to change the overall picture that many reptile species are present in a largely unregulated global trade. But if these analyses are included, I think they have to be accurate and communicated effectively. Alternatively, it could be that the paper is better served by cutting down some of the CITES and LEMIS material and focusing on the broader, species-level overview (as Reviewer #3 recommended in the last round of comments).

Response: Thank you for this, we are willing to take out most quantitative estimates; however, we feel highlighting that many species are wild sourced is also useful as it shows the conservation importance of better monitoring standards. We have tried to reduce the content on quantitative elements to just keypoints to enable a full understanding of what is

being traded and the significance. We have now carefully gone through the LEMIS data as you prescribe here, ensuring that we are quantifying individuals so we can be sure in statements and providing a breakdown for the different taxa (Figure S4) to highlight possible impacts on wild populations. We feel that by highlighting the number of species in trade (having ensured these are representative of individuals), and the origins it provides a unique and important insight into global trade dynamics and their impacts. We provide a fully detailed method and our entire analysis pipeline so that our drawn conclusions can be validated by other researchers, and that the results are an accurate representation of trade.

Please see my specific major and minor comments below.

Evan A. Eskew

Major Comments

Page 1, “The regulations primarily protect commercially traded, “charismatic” animal species; only recently covering lesser-known species (e.g. pangolins-2016).” I know this sentence has been subject to previous comment, but it still reads as inaccurate. It now seems as if you mean to suggest CITES *only* protects charismatic animal species, which is clearly untrue given that the majority of listed species are plants.

Response: We agree how the statement could have been misleading and have changed this to narrow toward animals because within the animal species particularly chordates there is a bias towards “charismatic” fauna. “Within animals the CITES regulations primarily regulate trade of commercially traded or “charismatic”...”

Page 2, Figure 1

I previously commented that it would be worth reviewing the unique reptile species names you’re using from LEMIS. There could very easily be minor misspellings or synonymies present. Your previous response partially addressed this issue by clarifying that taxonomies from all datasets were referenced against the Reptile Database species list. This addresses the concern that you could be double (or more) counting synonymous nomenclature for a single species. But it doesn’t entirely resolve the concern. If you haven’t thoroughly reviewed the unique LEMIS species names you’re analyzing, it’s possible that misspellings are resulting in undercounting (i.e., a species name in the database is a clear misspelling of a valid species name that doesn’t get matched against the Reptile Database and is therefore left out of your analysis). At the very least, I would give a qualifying statement about this, explaining the fact that these names, in their raw form, are not taxonomically standardized. Ideally, you review the LEMIS taxonomy to verify why any names are not matching the Reptile Database and resolve ambiguities. But at the very least, your readers should be clearly told this is an outstanding issue in the analysis.

Response: We have reviewed the failed matches, comparing the failed-to-match names against the species detected by other sources. We used the similars R package to find species names with fewer than 5 characters different. Of the failed matches only 27 “species” in LEMIS were not already reported in other data sources. We have added details on the overlap of the non-matched names to the methods.

We have added details on this quantification of mismatching in the methods:

“Outside of generic level listings, 83 names could not be matched. We compared the 83 names to the traded list from other sources, looking for names with fewer than 5 different characters (using the similars v.0.1.0 package [70]); 56 species were found to be present in other sources by this metric.”

Page 4, Figure 2C

I still find this figure panel unnecessarily confusing. Why not simply separate the unique species observed in online trade over time and the CITES species present in online trade over time?

These are completely distinct metrics with very different interpretations (one represents species that are not documented in any other year, the other does not). For this reason, I think the two sets of information would work better as two separate panels. At the very least, the y-axis labels should be more informative. Something like “Number of unique species in the online trade” and “CITES species present in the online trade”, respectively.

Response: We agree and have split the two elements of 2C into 2C and 2D. 2C is exclusively the unique species per year, and 2D the count of CITES species each year.

Page 4, Figure 2C

How could it be the case that a count of all names is ever less than strictly scientific names (as in the year 2007 here)? I think the distinction between these two categories deserves better explanation either here or in the Methods on page 13.

Response: There can be instances where the scientific only names produce more unique species for a given year. For example, let's say we only detected *Xenodermus javanicus* in 2007 and 2008, but the 2007 instance was via *Xenodermus javanicus* and the 2008 was via dragonsnake. In this case the number of species unique to 2007 would decrease when using all names because *Xenodermus javanicus* would have also been detected in 2008 via a common name. Essentially, using more keywords can lead to a decrease in the number of unique species in a given year because of the increased chances of detecting species in other years.

We have added some additional detail to the methods justifying why the two counts were included: “*To show the sensitivity to the keywords used, we counted the number of unique species in two ways: 1) counting all species detected using either scientific or common name keywords, 2) counting species only detected using scientific name keywords. The two keyword groups produce slightly different yearly species lists; therefore, changing the number of unique species per year and yearly residuals*”

Page 5, “In terms of traded items, LEMIS lists 63.0% (63.2% excluding seized shipments) as wild-sourced (456,722/724,655), nearly twice that listed as originating from captive, ranching, and commercial breeding (35.6% 258,021/724,655).”

If I understand correctly, you’ve just filtered the LEMIS data for class Reptilia, leaving 724,655 rows of data, which you treat as “items.” To me this is a major issue, and one that is directly analogous to Reviewer #3’s concerns about how you’ve treated CITES trade data in various iterations of the manuscript. Those rows of data you’re analyzing are not items. Rather they could be representative of multiple reptile-derived products. This distinction matters in the sense that it affects every trade metric you report for LEMIS or CITES data that does not rely on a gross, species-level analysis. To do these calculations accurately, they should actually be on the level of items, which will require more filtering of the data or a concentration on particular wildlife product descriptions (in the LEMIS data) or Terms (in the CITES data). For example, you might report the wild versus captive sourcing of only live animals or some specific reptile product like leather.

Response: Thanks for highlighting this. We have updated all LEMIS quantitative values focusing the analysis on items that represent individual animals (removing “parts” and

retaining only whole dead bodies, live eggs, dead specimens, live individuals, full specimens, substantially whole skins, and full animal trophies). We have removed the row-based summaries. We have also added the numbers pertaining only to live individual imports, thus the results now provide a much more accurate and comprehensive analysis of minimum numbers of individuals included.

We have added details in the methods that specify the LEMIS categories we included:

“To investigate the extent of wild capture in LEMIS data, we restricted our summaries to items that represent full animals (whole dead bodies, live eggs, dead specimens, live individuals, full specimens, substantially whole skins, and full animal trophies). Our quantification of non-commercial trade was calculated by the number of full animal items listed as Scientific, Reintroduction, or Biomedical research; our quantification of captive sourced trade was calculated by the number of full animal items listed as being bred/born in captivity, commercially bred, or from ranching operations. We excluded all instances of NA in either purpose or source filters. We summarised the quantity of traded items by genus, and further simplified the genus-summary to clade using Reptile Database genera and family information. For genera missing from Reptile Database (e.g., where genus information was family such as Varanidae), we manually assigned the clade.”

Page 8, Figure 4

I believe in the caption it should read, “The period 2000-2001 is only covered by LEMIS data.” I would also very, very explicitly mention that the LEMIS data has been filtered for commercial trade here (if indeed it has?). It’s not enough to bury this important information in the Methods.

Response: Thanks-LEMIS data was filtered to commercial trade only for time lag investigations. We have corrected the figure legend to make this clear.

Page 9, “...this totaled to over 63% of all imports.”

What exactly are you summarizing here? Wildlife product shipments? A sum of wildlife product items?

Response: Items representing whole individuals; this has now been added in text and detailed

Page 10, “For lower-value species banning trade from key-regions may not drive trade “underground” as can happen with higher value species (11). Conservationists actively hindered trade ban implementation for birds on such grounds (29), but when eventually applied within Europe global bird trade decreased by 90% (30).”

This would seem to be a very generous read of the Reino et al. paper. First, that paper documents a decrease in bird trade when the EU implemented a trade ban, but it also notes that global trade flows were reshaped by the ban. Second, you claim that reptiles are “lower-value” species, yet your paper mentions small-ranged, endemic species and the Courchamp et al. paper you also cite explicitly lays out how rarity might be inversely related to value. So I’m not sure it’s obvious that rare reptiles *would* be low value in a trade ban scenario. Finally, and most importantly, you’re relying on the perceived effectiveness of trade bans as suggested by the Reino et al. paper, yet that manuscript uses CITES trade data, which you claim throughout this manuscript is a rather limited perspective on wildlife trade. And it doesn’t address the potential for redirection to illegal trade channels, which is one of the main concerns with trade bans. All this is to say, I think you need a more nuanced discussion of potential policy solutions for global reptile trade that doesn’t exclusively push for potentially ineffective or harmful trade bans.

Response: Thanks, we have tried to provide a more nuanced and detailed discussion of this and include a number of other references to try to ensure we are representative and provide wider perspectives on the complexity of the issue. We note the reviewers perspectives on trade-bans and understand their perspectives based on their recent paper on the topic

[https://www.thelancet.com/journals/lanplh/article/PIIS2542-5196\(20\)30123-6/fulltext](https://www.thelancet.com/journals/lanplh/article/PIIS2542-5196(20)30123-6/fulltext)

Whilst we agree trade bans are a blunt instrument, banning the international trade of unassessed and endangered species without CITES appendix quotas seems a sensible alternative to a total ban, whilst enabling regulated trade of species that are unlikely to be severely impacted and can have their identity validated. Furthermore, trade requires monitoring, and monitoring and regulating wild-animal trade is something that from both a disease and conservation perspective should be a global standard, as it already is in the US (and Australia). We realise that increasing rarity can stimulate demand, but newly described species already have that issue as we demonstrate, yet these species lack even basic protection. Given that captive breeding should be a viable option within many of the major importing regions it is likely that international trade from the wild would not be required to meet demand for the majority of species. Furthermore, though as the reviewer points out CITES only gives a limited reflection on trade due to the subset of species monitored coupled with bans in major importing countries we feel that the Reino paper is a useful comparison, though we now more clearly highlight the challenges with inconsistently applied (regional) regulation.

We have rephrased the text as follows: *“Studies have demonstrated that CITES is consistently behind the IUCN in species assessment and inclusion (30). Until population impacts are known and assessments complete, trade-bans from specific regions, where exports include threatened species, should also be considered. For low-value species banning trade from key-regions may not drive trade “underground” as can happen with high-value species (11), especially when such actions are used to stimulate regulated markets based on captive breeding as was found effective with crocodiles (31). Conservationists actively hindered trade ban implementation for birds on such grounds (32), but when eventually applied within Europe global bird trade decreased by 90% (33), in part because of the availability of non-wild sourced alternatives for captive breeding (34). With just under 4000 reptile species found to be traded within this study there is ample stock already in captivity to justify the development of certified and monitored captive breeding within free-trade zones and prevent the need for commercial import. Such actions have already been used in the case of other taxa, for example “The Wild Bird Act” and “EU Wild bird ban” have prevented the importation of exotic birds into the US since 1992, and Europe since 2005 (<https://www.fws.gov/le/USStatutes/WBCA.pdf>). In both the cases of the US and Europe, disease risk and impact on native birds was the stated case for regional bans (35), and these same justifications exist in the case of reptiles (36, 37). Yet though undoubtedly effective in reducing global trade, unintended consequences, shifting routes and markets (34) also demonstrate a more holistic approach is needed such as a requirement for CITES listing for the export of reptiles, though species listed as Least Concern could be traded more widely if mechanisms for verification of species identity existed prior to export. From a standpoint of both disease and understanding trade systems like LEMIS should also become global standards for the export of live animals, and especially for wildlife export. Whilst many conservationists and organisations may challenge such an approach, as occurred in the case of birds (32), our data highlights thousands of species impacted by wild capture, including many which are new to science. By regulating what can rather than what cannot be traded internationally, we can considerably reduce the pressures on wild reptile populations.”*

Page 11, “We ceased cycling through search pages when a URL returned a 404 error, or when 100 pages had been cycled through. 100 pages were surveyed to prevent endless cycling back onto initial pages, or deriving errors from misinterpreting the number of search pages returned, whilst still exceeding the number of pages on most sites.”

Through your revisions you’ve convinced me that stopping at 100 pages has not led to any alphabetical bias in your coverage of online reptile trade (since sites list reptiles in different schema). But you still haven’t assuaged the concern that you’ve not completely characterized the stock list of every site. This doesn’t need to be redone necessarily, but I would explicitly state in the methods that your online data scraping strategy may result in undercounts of species in online trade given that there are reptile trade sites you may have incompletely assessed.

Response: We have added a sentence in the methods highlighting the possibility of undercounting species on larger websites, although we have caveated the statement by drawing attention to the high levels of overlap between websites. The species accumulation curves suggest that we are capturing the majority of the online trade (Fig. S1-S3), as the curves reach an asymptote.

We have added the following to the methods: “The 100 page limit may have led to missing species on large websites, but undercounting likely only affected a small portion of the websites searched via cycling methods and overlap between websites species lists mitigate suboptimal sampling on any particular website (see species accumulation curves Fig. S1-S2).”

Page 13, “We counted the unique species in two ways: using all names to detect species, and only scientific names.”

As mentioned in my comment on Figure 2C, I think the distinction/definitions here need much more precise explanation so that readers can better interpret your results.

Response: We have added details to the methods to better differentiate the two keyword approaches. See above response to comments on Page 4, Fig. 2C.

“To show the sensitivity to the keywords used, we counted the number of unique species in two ways: 1) counting all species detected using either scientific or common name keywords, 2) counting species only detected using scientific name keywords. The two keyword groups produce slightly different yearly species lists; therefore, changing the number of unique species per year and yearly residuals”

Page 14, “For LEMIS species counts we included those only listed to genus level, for example *Anolis sp.* would be counted as a species alongside *Anolis carolinensis* and *Anolis smaragdinus* etc. As for CITES species lists, the unstandardised LEMIS names were matched to those present in Reptile Database (operating as our backbone nomenclature), leading to both synonymisations and splits. A LEMIS name was converted to a Reptile Database name if it matched any current, common, or historically used name. By LEMIS naming there were 639 instances of genus level listing, that were matched to 510 Reptile Database names. Of the 510 converted names, 442 appeared in other sources, suggesting genus level listings in LEMIS did not inflate species counts. Those that failed to be converted were not included in total species counts; as final counts were entirely based on Reptile Database explicit species naming. Final species counts from all data sources are based on unique Reptile Database names, after this synonymization/split process.”

I would be very specific here in the final sentence and state that your final species counts do not include any generic identifiers (if indeed they don't). The discussion here of the number of genus-level listings in LEMIS is almost a distraction from the issue because I thought, from your response to reviewers, that genus-level IDs don't make it into the species counts (shown in Figure 1) at all. But the phrase here "suggesting genus level listings in LEMIS did not inflate species counts" seems to state that the genus-level IDs in LEMIS matched with other genus-level IDs in other data sources (that were included in the final species counts)?

Response: We have clarified that any listings remaining at a generic level listings were excluded. We have retained the details on matching generic-level listings to Reptile Database names, because this reflects exactly what our code did, and shows that genus-level ambiguity was not a significant issue because of data source overlap. We filtered out generic names after Reptile Database matching, largely to ensure that genera were not being missed as a cross-reference to our number of species and genera.

We have removed a sentence that could be interpreted as stating generic level listings were included in the final count, it was meant to state that the generic level listings were initially dealt with like full names (ie. subject to the Reptile Database conversion) to check completeness, but not included as species in the final lists or subsequent analysis.

Page 14, "For CITES we used the summary statistics tool in ArxMap to quantify the means and totals for the numbers exported and imported..."

Yet earlier on this page you only mention gathering CITES export data? I think this is important because this ambiguity links in with many of the concerns Reviewer #3 has had throughout the review process: with the CITES data, it's very important that you be clear which dataset you're working with and how you are reporting trade statistics (i.e., importer- or exporter-reported data).

Response: Apologies for the ambiguity. We list imported and exported values for CITES, and only give one figure where the two are the same, we have now corrected the figures and terms accordingly to ensure what is being quoted is clear throughout. We have also corrected the methods to note that all trade was downloaded from comparative tabulations.

Minor Comments

Page 1, "Here we highlight the scope of the global reptile trade: over 36% of reptile species are traded largely for pets, over three quarters of which are not covered by international trade regulation, including numerous endangered or range-restricted species, with hotspots in Asia."

This is a bit of a sprawling, hard-to-follow sentence. Given that it contains some of your paper's key results, I would consider revising for simplicity.

Response: this line has been split into two and simplified

"Here we highlight the scope of the global reptile trade: which includes over 36% of reptile species, over three quarters of which are not covered by international trade regulation. These species include numerous endangered or range-restricted species, with hotspots in Asia."

Page 2, "Only CITES and LEMIS trade databases include a percentage of seized specimens (LEMIS, 14969/724656 2.07% of individuals were categorised as seized, CITES 0.2-0.4% overall (imported 781421/197819654; exported 425156/206360017))."

I would consider revising the punctuation in this sentence for clarity.

Response: Thanks, clarified “(LEMIS; 2% seized and CITES 0.2/0.4% (imports/exports)). Outside these small percentages, other trade is legal, or purporting to be legal in the case of online trade”

Page 3, “...2.18% of individuals listed in LEMIS are for non-commercial purposes...”
Is this meant to imply live individuals?

Response: The text formerly referred to all items, we now include items that represent individuals (whole dead bodies, live eggs, dead specimens, live individuals, full specimens, substantially whole skins, and full animal trophies) only and found that 1.54% of individuals are for non-commercial trade. We have updated the values in the manuscript to show this new more restricted items-representing-individuals approach.

Page 3, “This contrasts to the species reported by CITES trade database, where 90% of species are listed in trade are in a CITES appendix...”

One too many “are”s here.

Response: Removed and rephrased to “*This contrasts to the CITES trade database, where 90% of species listed in trade are in a CITES appendix (e.g appendix-2: 72.6% 613/844; appendix-1 11% 94/844).*”

Page 3, “Not all CITES trade database species are covered by CITES appendices. Species reported in the CITES database may come from seizures (2.8-4%), or from shipments which include both species with CITES appendices, and unlisted species.”

I don’t think this is a completely accurate description of the way these data arise. It’s not simply the case that unlisted species are randomly caught up in the CITES reporting process because of their co-occurrence with CITES-listed species. Rather, there are other mechanisms for non-CITES-listed species to be reported in this database, including the EU Wildlife Trade Regulations (https://ec.europa.eu/environment/cites/species_en.htm).

Response: We have added “in addition to other mechanisms”

Page 3, “Listings in LEMIS covered large volume imports to fashion companies (which are listed), as well as much smaller numbers of a diverse selection of species to other buyers.”

What does the “which are listed” phrase refer to? Listed on what?

Response: LEMIS data lists the importers, we have added the phrase “*importers are listed in the LEMIS database for each shipment*”

Page 3, “Raw counts of species indicate the consistent presence of CITES protected species in the trade (75.5 ±5.81; Fig. 2C).”

In which trade? I believe it’s critical to point out this is online trade, correct?

Response: Yes, figure 2C is only the online trade. We have added this detail in the sentence to reflect the figure caption.

Page 5, “In addition all 13 species imported into over 125 countries are crocodiles, pythons and monitors, and the three species exported from 125 countries or more include two crocodiles and one python species.”

This is all a bit difficult to follow. I assume you’re talking about the 13 species in the CITES trade database that happen to fall within the five genera you previously mentioned?

Response: This has been rephrased to clarify “On a global basis 13 species were imported into over 125 countries, which were all crocodiles, pythons and monitors, and three species (two crocodiles and one python species) were exported from 125 countries.”

Page 5, “The commercial focus of CITES is further reflected in the regulation of fashion targeted species: 100% of crocodiles and 52% of Testudines have a CITES appendix, compared with 9% of lizards and 4% of snakes have CITES appendix listings, which coincides with those used commercially in fashion, with 20 of the 22 species of crocodilian listed with commercial or personal “purposes”.”

This remains unclear to me. Do you mean to say that 100% of all crocodiles and 52% of all Testudines species globally are covered by CITES? Are the 22 crocodilian species only those listed by CITES or are you saying that there are 22 crocodilian species globally? Are the 20 species *exclusively listed* with commercial or personal purposes or those that were *ever listed* with commercial or personal purposes?

Response: This has been clarified “The commercial focus of CITES is further reflected in the regulation of fashion targeted species: 100% of crocodiles and 52% of Testudines classified globally have a CITES appendix, compared with only 9% of lizard species and 4% of snakes. This coincides with species used commercially in fashion, with 20 of the 22 species of crocodilian listed in trade with commercial or personal “purposes”.”

Page 5, “In total 53% of CITES reptile items traded are from wild caught animals (whereas 33-36% are from captivity).”

I think for the CITES data, it’s always critical to mention whether you’re discussing importer- or exporter-reported data.

Response: Thank you, this has been clarified in text, when one percentage is given then it was the same for both, whereas two represents the range

Page 5, “The purpose is largely listed as commercial (95-96%%), with a minority for personal use (0.1%), highlighting that within CITES monitored data the majority is for commercial (largely fashion) purposes, and neglects trade for other purposes such as the pet-trade.”

This phrasing almost makes it sound like CITES is making an active decision to neglect pet trade. I would consider saying that pet trade is not “widely represented” in the CITES trade database or something similar.

Response: Thank you, we replaced “neglects” with “may overlook”

Page 5, “Data from LEMIS shows that for 92% of species have wild-caught individuals imported...”

Don’t need the “for”

Response: removed

Page 5, “...has more than 50% of their reptile species covered by CITES regulations...”

Do you mean their *traded* reptile species?

Response: Thanks, corrected

Page 9, “...those for sale online or in LEMIS.”

This could be read ambiguously. Species aren’t for sale in LEMIS. Their trade is documented by

LEMIS.

Response: Thanks, changed to “documented as traded via”

Page 9

I believe “Lacey act” should appear as “Lacey Act” here and throughout the manuscript.

Response: Thanks, corrected

Page 11, “...to retrieve a list of reptiles selling websites...”

Should be “reptile selling websites”

Response: Thanks, corrected. Page 12, “Five CITES listed species had no matching counterpart in the Reptile Database, we determined this was caused by minor spelling mistakes.”

Comma splice.

Response: Thanks, corrected. We have replaced the comma with a semicolon.

Page 23, Figure S3

Your response to my previous concerns about this Figure was a bit confusing. My comment led you to revise text in other parts of the manuscript, but the problem with this Figure remains: your caption states the data considered is from 2004-2018 (15 years) whereas the sample coverage value does not intersect the x-axis at year 15 but rather at some greater year value (I can’t tell which exactly). The response to reviewers was also confusing because it suggested that the years 2002-2019 would constitute 17 years of data when it’s actually 18.

Response: Apologies, the caption for the figure was incorrect, it should have read 2002-2019, reflecting statements in the methods. You are correct, it should be 18 data points, as both 2002 and 2019 are included. We have added the sample size for both curves in the methods.

Reporting Summary, “Research sample” section

It may not be completely accurate to say that the subset of LEMIS data you analyzed represents legal trade (if some of the data are in fact from seized shipments).

Response: We have added in the fact that a small % of LEMIS traded data represents seized goods. “LEMIS trade database details the animal imports into the USA, these records are known declared trade events, with a small minority detailing seized imports.”

Reviewers' Comments:

Reviewer #1:

Remarks to the Author:

Following the last round of revision, I find there have been major improvements to the manuscript. Methods throughout are better described, including how scientific names were matched and counted among the various data sources. In addition, the changes to Figure 2 result in a clearer data presentation. Finally, I'm now much more comfortable with the way the LEMIS data is analyzed as the authors have appropriately narrowed down the diverse data represented there to approximate numbers of trade reptiles (and these analytical choices are described in the methods). In sum, these changes bolster the scientific conclusions of the manuscript and should increase its utility to the research community. In terms of the science, I think the article is now ready for publication.

However, I would encourage the authors to read through the article with fine attention to detail as they undergo final preparations for publication. I did not record every instance in my comments here, but there were some minor outstanding grammatical and stylistic issues (missing commas, missing end parentheses, etc.) that should be corrected prior to publication.

Although not directly related to the manuscript, I would also encourage the authors to think through their vision for global reptile trade policy prior to publication. The Discussion section of the manuscript is compelling and informative as is, but I suspect the implications of this article's findings for reptile trade policy will receive significant attention on social media and in policy circles. The shift in status quo that the authors endorse would obviously entail some major regulatory changes, and there are a host of related questions to answer. For example, which organization or regulatory body would be in charge of conducting the assessments that clear or deny a reptile species for inclusion in global trade? The IUCN? Somebody else? What sort of global data monitoring would assist scientists and policymakers in these efforts (and how similar to a system like LEMIS would it be)? I don't pretend to have a comprehensive understanding of all the issues involved, and the authors shouldn't be expected to either, but I do think they will be looked to as experts on these issues following the release of the article. So they are definitely questions worth thinking more about.

Best of luck with the publication of this manuscript and the broader science communication efforts that will likely accompany it.

Evan A. Eskew

Major Comments

It appears that the article references have been thrown off at some point during the revision process and should be thoroughly reviewed prior to publication. For example, the reference for the *lemis* R package is not ref. 70, the reference for the *similiars* R Package is not ref. 71, etc.

Page 5: "The commercial focus of CITES is further reflected in the regulation of fashion targeted species: 100% of crocodiles and 52% of Testudines classified globally have a CITES appendix,

compared with only 9% of lizard species and 4% of snakes. This coincides with species used commercially in fashion, with 20 of the 22 species of crocodilian listed in trade with commercial or personal “purposes”.

While many instances of ambiguity from the previous manuscript draft have been adequately resolved, some, like in this section, remain. In finalizing the manuscript, I would encourage the authors to revise with particular attention to whether there's any ambiguity about which data source (online, LEMIS, CITES, IUCN) is being referenced at any given time. For example, in this passage, the "classified globally" refers to which organization? The IUCN I'm assuming, but it becomes difficult for a reader to follow your cross-referencing of one dataset with another if you do not give them the explicit verbal sign posts. The 20 of the 22 species of crocodilian are "listed in trade" by whom? CITES, right? Such small clarifications will go a long way. If ambiguity remains in the final manuscript, I would expect that readers will contact you for clarification or there may be response articles targeting issues of data interpretation and communication.

Page 8: "...while the other 41 species were only detected in the 2019 snapshot data so their initial date of appearance in the trade is unknown (Fig. 4B)."

You may want just another phrase or sentence to explain this a bit more. I assume you're thinking that for many of those 2019 snapshot species you might actually have detected them in the trade earlier if you conducted a temporal analysis with a broader selection of websites?

Minor Comments

Page 2: "This online trade represents one of the interfaces between buyers and sellers, but the origin of these individuals is not declared due to a lack of regulations requiring such details, thus in addition to considering online trade findings we additionally explored two major international trade databases."

Run-on sentence.

Page 2: "Both CITES and LEMIS trade databases include a percentage of seized items, but like online trade the majority of all items are legally traded (99%) and under 1% is from illegal trade (LEMIS; 0.2% of individuals are seized and in CITES 0.2/0.4% (imports/exports))."

Specify that these "individuals" are individual reptiles? In addition, I don't know that the punctuation changes here helped clarify completely. In similar situations, I would use the semicolon to separate information about the two different data sources (LEMIS and CITES).

Page 4, Figure 2 caption: "Trend in number species detected on the most species rich reptile trading website."

Missing an "of".

Page 5: "In total 53% of CITES reptile items traded are from wild caught animals (whereas 33/36% are from captivity, this remains consistent if filtered to represent individuals at 48/47% wild (I/E) and 34/39% captive bred)."

I know this has been revised from previous versions and is explained in the methods, but I think it will still read as ambiguous for readers going through the article for the first time (who won't have encountered the methods yet). Why not just state explicitly that 53% of CITES reptile items are wild caught whether using importer- or exporter-reported data? Why not define "(I/E)" explicitly as "(Importer/Exporter reported data)" since it's your first time ever using this

abbreviation? Why not introduce the Importer/Exporter designation immediately after the “33/36%” since that designation applies to those data, and they’re presented first in this series?

Page 7, Figure 3B: The negative symbols (or what appear to be negative symbols) in the figure legend could be confusing. I would just remove them entirely.

Page 8: “...11 newly described species were detected in the trade by the next year.”
May be better to explicitly say: “were detected in the trade by the year following description.”

Page 10: “Our findings suggest a minimum of 36% reptile species...”
Missing a second “of” in this sentence.

Page 11: I previously mentioned inconsistency in the rendering of “Lacey Act” throughout the manuscript, but now it still appears variously as “Lacey Act,” “Lacey act,” and “Lacey-act.”

Page 17: “79,796,472 items including skulls and skeletons; 79,812,310 excluding skulls and skeletons”

Are these two numbers flipped? It seems the “including” figure should be larger than the “excluding” figure.

Page 17: “To obtain overall number of species and percentage species...”
“Percentage of species”?

Reviewer Letter:

Following the last round of revision, I find there have been major improvements to the manuscript. Methods throughout are better described, including how scientific names were matched and counted among the various data sources. In addition, the changes to Figure 2 result in a clearer data presentation. Finally, I'm now much more comfortable with the way the LEMIS data is analyzed as the authors have appropriately narrowed down the diverse data represented there to approximate numbers of trade reptiles (and these analytical choices are described in the methods). In sum, these changes bolster the scientific conclusions of the manuscript and should increase its utility to the research community. In terms of the science, I think the article is now ready for publication.

Response: Thank you, we are glad to fully satisfy you with our methodology and analysis, and appreciate all your efforts to bolster the value of the paper.

However, I would encourage the authors to read through the article with fine attention to detail as they undergo final preparations for publication. I did not record every instance in my comments here, but there were some minor outstanding grammatical and stylistic issues (missing commas, missing end parentheses, etc.) that should be corrected prior to publication.

Response: Thank you, all three authors have now fully proofed the manuscript to ensure grammar is correct throughout

Although not directly related to the manuscript, I would also encourage the authors to think through their vision for global reptile trade policy prior to publication. The Discussion section of the manuscript is compelling and informative as is, but I suspect the implications of this article's findings for reptile trade policy will receive significant attention on social media and in policy circles. The shift in status quo that the authors endorse would obviously entail some major regulatory changes, and there are a host of related questions to answer. For example, which organization or regulatory body would be in charge of conducting the assessments that clear or deny a reptile species for inclusion in global trade? The IUCN? Somebody else? What sort of global data monitoring would assist scientists and policymakers in these efforts (and how similar to a system like LEMIS would it be)? I don't pretend to have a comprehensive understanding of all the issues involved, and the authors shouldn't be expected to either, but I do think they will be looked to as experts on these issues following the release of the article. So they are definitely questions worth thinking more about.

Response: Thank you. The paper should be out in good time for the IUCN and CBD meetings next year, and CITES is discussing changing its approaches, and from my last discussion with John Scanlon CITES would remain responsible for international wildlife trade, even if the scope and mission changed from what it was originally set up to do. Building on this paper we are now starting a policy paper to go into more depth on options for regulation, and working in concert with individuals in those institutions to ensure such options would be viable.

Best of luck with the publication of this manuscript and the broader science communication efforts that will likely accompany it.

Evan A. Eskew

Response: Thanks again for working with us to improve the manuscript and enable us to provide all the necessary detail to ensure readers could be fully confident in our results. We really do appreciate the time and attention you have given to improving all aspects of the paper.

Major Comments

R1: It appears that the article references have been thrown off at some point during the revision process and should be thoroughly reviewed prior to publication. For example, the reference for the lemism R package is not ref. 70, the reference for the similars R Package is not ref. 71, etc.

Response: Thanks, all references have now been checked and corrected.

R1: Page 5: “The commercial focus of CITES is further reflected in the regulation of fashion targeted species: 100% of crocodiles and 52% of Testudines classified globally have a CITES appendix, compared with only 9% of lizard species and 4% of snakes. This coincides with species used commercially in fashion, with 20 of the 22 species of crocodylian listed in trade with commercial or personal “purposes”.”

Response: Edited slightly, now reads:

"The commercial focus of CITES is further reflected in the regulation of fashion targeted species: 100% of crocodiles and 52% of Testudines described globally have a CITES appendix, compared with only 9% of lizard species and 4% of snakes. This coincides with species used commercially in fashion, with 20 of the 22 species of crocodylian in trade in the CITES database listed with commercial or personal 'purposes'"

R1: While many instances of ambiguity from the previous manuscript draft have been adequately resolved, some, like in this section, remain. In finalizing the manuscript, I would encourage the authors to revise with particular attention to whether there's any ambiguity about which datasource (online, LEMIS, CITES, IUCN) is being referenced at any given time. For example, in this passage, the “classified globally” refers to which organization? The IUCN I'm assuming, but it becomes difficult for a reader to follow your cross-referencing of one dataset with another if you do not give them the explicit verbal signposts. The 20 of the 22 species of crocodylian are “listed in trade” by whom? CITES, right? Such small clarifications will go a long way. If ambiguity remains in the final manuscript, I would expect that readers will contact you for clarification or there may be response articles targeting issues of data interpretation and Communication.

Response: Thanks, a good point. We have now gone through again to ensure everything is entirely clear.

R1: Page 8: “...while the other 41 species were only detected in the 2019 snapshot data so their initial date of appearance in the trade is unknown (Fig. 4B).” You may want just another phrase or sentence to explain this a bit more. I assume you're thinking that for many of those 2019 snapshot species you might actually have detected them in the trade earlier if you conducted a temporal analysis with a broader selection of websites?

Response: Thanks, clarified and added the likely direction of the impact if we have treated 2019 the same as differently sampled years:

“The unequal sampling between 2019 and previous years led us to treat species only detected in the 2019 snapshot separately. Species only detected in 2019 likely have an earlier initial date of appearance (missed due differences in sampling methods); including lag times based on 2019 detections would have biased the mean upwards.”

Minor Comments

R1: Page 2: “This online trade represents one of the interfaces between buyers and sellers, but the origin of these individuals is not declared due to a lack of regulations requiring such details, thus in addition to considering online trade findings we additionally explored two major international trade databases.” Run-on sentence.

Response: Split into two sentences for clarity

R1: Page 2: “Both CITES and LEMIS trade databases include a percentage of seized items, but like online trade the majority of all items are legally traded (99%) and under 1% is from illegal trade (LEMIS; 0.2% of individuals are seized and in CITES 0.2/0.4% (imports/exports)).”

Specify that these “individuals” are individual reptiles? In addition, I don’t know that the punctuation changes here helped clarify completely. In similar situations, I would use the semicolon to separate information about the two different data sources (LEMIS and CITES).

Response: changed to individual reptiles, and semicolon added

R1: Page 4, Figure 2 caption: “Trend in number species detected on the most species rich reptile trading website.” Missing an “of”.

Response: Thanks, corrected

R1: Page 5: “In total 53% of CITES reptile items traded are from wild caught animals (whereas 33/36% are from captivity, this remains consistent if filtered to represent individuals at 48/47% wild (I/E) and 34/39% captive bred).” I know this has been revised from previous versions and is explained in the methods, but I think it will still read as ambiguous for readers going through the article for the first time (who won’t have encountered the methods yet). Why not just state explicitly that 53% of CITES reptile items are wild caught whether using importer- or exporter-reported data? Why not define “(I/E)” explicitly as “(Importer/Exporter reported data)” since it’s your first time ever using this abbreviation? Why not introduce the Importer/Exporter designation immediately after the “33/36%” since that designation applies to those data, and they’re presented first in this series?

Response: Thanks, we now define I/E at first use to ensure it is fully comprehensible to the reader

R1: Page 7, Figure 3B: The negative symbols (or what appear to be negative symbols) in the figure legend could be confusing. I would just remove them entirely.

Response: Not negative, just a formatting issue, fixed

R1: Page 8: “...11 newly described species were detected in the trade by the next year.”

May be better to explicitly say: “were detected in the trade by the year following description.”

Response: Thanks, text has been changed as suggested

R1: Page 10: “Our findings suggest a minimum of 36% reptile species...”
Missing a second “of” in this sentence.

Response: Thanks, added

R1: Page 11: I previously mentioned inconsistency in the rendering of “Lacey Act” throughout the manuscript, but now it still appears variously as “Lacey Act,” “Lacey act,” and “Lacey-act.”

Response: Thanks, we have now changed this to “Lacey Act” throughout to match the listing on official documents

R1: Page 17: “79,796,472 items including skulls and skeletons; 79,812,310 excluding skulls and Skeletons” Are these two numbers flipped? It seems the “including” figure should be larger than the “excluding” figure.

Response: Thank you for spotting that, yes exactly that, the figures were flipped, and have now been corrected.

R1: Page 17: “To obtain overall number of species and percentage species...” “Percentage of species”?

Response: Yes, thanks-corrected